# In-Context Learning of Linear Dynamical Systems with Transformers: Approximation Bounds and Depth-separation

**Frank Cole**[*]
School of Mathematics
University of Minnesota, Twin Cities
Minneapolis, MN 55455
cole0932@umn.edu

**Yuxuan Zhao**[*]
School of Mathematics
University of Minnesota, Twin Cities
Minneapolis, MN 55455
zhao1895@umn.edu

**Yulong Lu**
School of Mathematics
University of Minnesota, Twin Cities
Minneapolis, MN 55455
yulonglu@umn.edu

**Tianhao Zhang**
School of Mathematics
University of Minnesota, Twin Cities
Minneapolis, MN 55455
zhan7594@umn.edu

## Abstract

This paper investigates approximation-theoretic aspects of the in-context learning capability of the transformers in representing a family of noisy linear dynamical systems. Our first theoretical result establishes an upper bound on the approximation error of multi-layer transformers with respect to an $L^2$-testing loss uniformly defined across tasks. This result demonstrates that transformers with logarithmic depth can achieve error bounds comparable with those of the least-squares estimator. In contrast, our second result establishes a non-diminishing lower bound on the approximation error for a class of single-layer linear transformers, which suggests a depth-separation phenomenon for transformers in the in-context learning of dynamical systems. Moreover, this second result uncovers a critical distinction in the approximation power of single-layer linear transformers when learning from IID versus non-IID data.

## 1 Introduction

Transformers Vaswani [2017] have achieved remarkable success in natural language processing, driving the success of modern large language models such as ChatGPT Achiam et al. [2023]. The impressive capabilities of transformers in NLP tasks has spurred their adoption across diverse domains beyond NLP, such as computer vision Li et al. [2023], Khan et al. [2022], Chen et al. [2020], Ramesh et al. [2021], computational biology Jumper et al. [2021], Choi and Lee [2023], physical modeling Batatia et al. [2023], McCabe et al. [2023], Subramanian et al. [2024], Ye et al. [2024], among others. A particularly intriguing property of transformers is their ability to perform *in-context learning*: pre-trained transformers are able to make accurate predictions on unseen sequences, even those beyond the support of their pre-training distribution, without any parameter updates. One promising explanation for these striking behaviors is the *mesa-optimization hypothesis* Von Oswald et al. [2023a,b], which says that transformers make next-token predictions by implicitly optimizing a context-dependent loss according some optimization algorithm encoded during pre-training.

---

[*]Equal contribution.

39th Conference on Neural Information Processing Systems (NeurIPS 2025).

To elucidate these emergent properties, a lot of recent research has studied in-context learning in tractable settings. For instance, several works studied in-context learning of linear models by simplified transformer architectures Zhang et al. [2023], Mahankali et al. [2023], Ahn et al. [2023]. In this setting, it has been shown that the transformer which minimizes a natural $L^2$ population risk indeed performs in-context learning over the class of linear models, and its prediction error decays at the parametric rate $O\left(1/T\right)$, where $T$ is the number of samples of the downstream task. Moreover, these trained transformers enjoy prediction error bounds that hold *uniformly* over the set of admissible tasks, despite only being trained on an $L^2$ loss. The learning algorithm encoded by transformers trained on in-context examples of linear regression can be expressed as a single step of gradient descent on a context-dependent loss, confirming earlier hypotheses by empirical works Von Oswald et al. [2023a]. Most of the prior theoretical works described in the previous paragraph works assume IID data, where each sequence $(x_0, x_1, \ldots, x_T)$ consists of uncorrelated tokens. A more realistic assumption is that the data/tokens are correlated, and linear dynamical systems provide a natural and tractable setting to study in-context learning of correlated data. In this non-IID setting, many theoretical questions on the in-context learning capability of transformers have remained elusive:

1. Given a class of dynamical systems, does there exist a transformer which in-context learns the class? If so, can we quantify the prediction error rate achieved by the transformer?

2. How complex must the transformer architecture be in order to perform in-context learning?

3. What algorithm do transformers trained to perform in-context learning over dynamical systems encode?

In this work, we focus on the first two questions above and investigate the approximation power of linear transformers in in-context learning for a class of linear dynamical systems which are corrupted by noise. Specifically, we are interested in learning from sequences $(x_0, x_1, \ldots, x_T)$ that are generated according to the linear stochastic dynamics

$$x_t = W x_{t-1} + \xi_t,\ 1 \le t \le T, \tag{1}$$

where $W \in \mathbb{R}^{d \times d}$ is a matrix whose eigenvalues lie in $(0, 1)$ and $\xi_t$ is independent Gaussian noise. Our goal is to derive quantitative results that characterize the ability (or inability) of linear transformers to learn the linear map defined by the conditional expectation $x_{t-1} \mapsto \mathbb{E}[x_t | x_{t-1}]$ in-context, from dynamical systems governed by Equation (2). Our main contributions are highlighted as follows.

1. We establish an approximation error bound for a class of deep linear transformers in learning the family of dynamical systems (2) in-context; see Theorem 1. More precisely, we show that linear transformers whose depth scales as $O(\log(T))$ can achieve an error of $O\left(\log(T)/T\right)$ with respect to an $L^2$-testing loss uniformly defined across tasks. The proof is based on constructing a transformer which approximates the least-squares estimator of the system (2) through iterative methods, while leveraging statistical properties of the least-squares estimator.

2. We further investigate the role of depth of transformers in our first result. We demonstrate that a broad class of single-layer linear transformers is fundamentally limited in its ability to perform in-context learning for the system described by Equation (2). Specifically, the test loss satisfies a lower bound that remains independent of the transformer's parameterization, even as the length of the trajectory data $T$ approaches infinity; see Theorem 2.

To the best of our knowledge, our work is the first to quantitatively analyze the in-context population loss for non-IID data, and the techniques required differ substantially from the IID case. In particular, our second result uncovers a distinct feature of single-layer linear transformers: their inability to learn from dependent data. This was observed in a different setting in Zheng et al. [2024]. In contrast, it has been shown in the IID setting that (see e.g. Zhang et al. [2023]) single-layer linear transformers *provably* learns linear models in-context, in the sense that optimized transformer can achieve zero test loss in the infinite context-length limit. We will expand upon the reasons behind the differing outcomes between these two settings in more detail in Section 3.3.

## 1.1 Related work

**ICL in the IID setting:** In the setting of IID-data, Garg et al. [2022] introduced the concept of in-context learning a function class and demonstrated experimentally that trained transformers

can indeed learn simple function classes in-context. Several empirical works provided evidence that transformers perform in-context learning by implementing a mesa-optimization algorithm to minimize a context-dependent loss Von Oswald et al. [2023a], Akyürek et al. [2022], Dai et al. [2022], Müller et al. [2021]. Subsequently, several theoretical works proved that single-layer linear transformers indeed implement one step of gradient descent to solve linear regression tasks Zhang et al. [2023], Mahankali et al. [2023], Ahn et al. [2023], Wu et al. [2023]. Recent works have refined the analysis of single-layer linear transformers, proving additional results on expressivity Zhang et al. [2024], adaptivity Vladymyrov et al. [2024], adversarial robustness Anwar et al. [2024], and task diversity Cole et al. [2024], Lu et al. [2024]. For softmax attention, recent works Collins et al. [2024], Li et al. [2024] derived in-context learning guarantees for one-layer transformers and related their inference-time prediction to nearest neighbor algorithms. Concerning deep models, the theoretical works Giannou et al. [2024], Fu et al. [2023] proved that multi-layer transformers can implement higher-order optimizations such as Newton's method. Beyond linear functions, several works have proven approximation, statistical, and optimization error guarantees for in-context learning over larger (possibly nonparametric) classes of nonlinear functions Yang et al. [2024], Guo et al. [2023], Kim and Suzuki [2024], Bai et al. [2024], Kim et al. [2024], Li et al. [2025].

**ICL in the non-IID setting:**   There are substantially fewer theoretical works on the in-context learning capabilities of transformers for non-IID data. Several works have analyzed transformers in the context of state space modeling and filtering problems Goel and Bartlett [2024], Li et al. [2023], Akram and Vikalo [2024], Ziemann et al. [2024]. Closer to our setting, the work Von Oswald et al. [2023b] extended the constructions of Von Oswald et al. [2023a] to the autoregressive setting and hypothesized that autoregressively-trained transformers learn to implement mesa-optimization algorithms to solve downstream tasks. The works Sander et al. [2024], Zheng et al. [2024] share a similar setting to our work and they prove that the construction in Von Oswald et al. [2023b] is optimal for data arising from a class of noiseless linear dynamical systems; in particular, Zheng et al. [2024] provides a sufficient condition under which the mesa-optimizer is learned during pre-training. Huang et al. [2024] studied the convergence of trained transformers on a different non-IID next-token prediction task. The work Ziemann et al. [2024] proved a lower bound on the number of parameters required to represent non-ergodic state space models in a different setting from ours.

Notably, the work Guo et al. [2023] and the concurrent work Wu et al. [2025] study the ability of transformers to approximate least squares algorithms for predicting (possibly more complex) dynamical systems. Our work differs from these two in that, while we study a simpler linear dynamical system, we prove approximation bounds with respect to the next token prediction error on noisy data, whereas the previous works only consider the ability of transformers to implement the least squares algorithm. In particular, our results requires us to synthesize our particular construction with statistical guarantees of the least-squares estimator (LSE) on noisy data, and to control the error incurred on the event where the LSE bounds fail to hold. Additionally, our constructions for the upper bounds are more parameter efficient (by leveraging the Richardson algorithm as opposed to gradient descent) and our lower bounds are novel to the best of our knowledge.

**ICL of Markov chains:**   A different line of work analyzes the ability of transformers to perform in-context learning over Markov chains on a finite state space. The work Nichani et al. [2024] studied this problem for two-layer softmax transformers trained by gradient descent and found that the model learns to predict the next token by inductively reading all previous instances of the current token. This mechanism was referred to as 'induction heads' by an earlier empirical work Olsson et al. [2022]. Several recent works have provided additional theoretical and numerical evidence to suggest that transformers learn to implement induction heads Edelman et al. [2024], Rajaraman et al. [2024], Chen et al. [2024]. Our problem setting can be viewed as a Markov chain on an infinite state space and is thus beyond the scope of the aforementioned works.

## 1.2   Organization

The paper is organized as follows. In Section 2, we discuss the details of our problem setup, including the observation model, the transformer architecture, and the loss functions considered herein. In Sections 3.1 and 3.2, we state our theoretical results on the approximation of deep and single-layer transformers respectively. We also discuss the qualitative difference we uncover between in-context learning with IID and non-IID data in Section 3.3. In Section 5 we sketch the proofs of our main

theorems, and in Section 6 we describe some directions for future work. All proofs are deferred to the appendix; Appendix A provides the proofs for Section 3.1, Appendix B provides the proofs for Section 3.2, and Appendix C provides the proofs for various auxiliary lemmas.

## 2 Set-up

### 2.1 The observation model

We consider in-context learning of sequences $\mathbf{x} = (x_0, x_1, \ldots, x_T) \subset \mathbb{R}^d$ defined by the noisy linear dynamical system

$$\begin{cases} x_0 = 0, \\ x_t = W x_{t-1} + \xi_t, \ 1 \le t \le T, \end{cases} \tag{2}$$

where $W \in \mathcal{W} \subseteq \mathbb{R}^{d \times d}$ and $\{\xi_1, \ldots, \xi_t\}$ are independent noise terms which follow a Gaussian distribution $N(0, \sigma^2 \mathbf{I}_d)$ for some $\sigma > 0$ which describes the signal-to-noise ratio of the tasks. We let $p_W$ denote a probability measure supported on $\mathcal{W}$. A sequence $\mathbf{x}$ is then generated according to the two-stage procedure where we 1) sample a task $W \sim p_W$, and then 2) conditioned on $W$, sample $\mathbf{x}$ according to the observation process (2). Our theoretical results are agnostic to the choice of the task distribution $p_W$ and depend only on its support $\mathcal{W}$. To make the analysis more tractable, we place the following assumptions on $\mathcal{W}$.

**Assumption 1.** *There exist constants $0 < w_{\min} < w_{\max} < 1$ such that*

$$w_{\min} \cdot \mathbf{I}_d \prec W \prec w_{\max} \cdot \mathbf{I}_d, \quad \forall W \in \mathcal{W}.$$

The assumption that the eigenvalues of the matrices in $\mathcal{W}$ are strictly less than one ensures that the dynamical system defined in Equation (2) is geometrically ergodic, which guarantees the loss functions to be considered have finite limits as $T$ tends to infinity. In contrast, previous works assume that the underlying dynamical system is noiseless, and therefore require that the task matrices have unit norm (e.g., rotation matrices) in order for the dynamics to be non-degenerate; see e.g. Sander et al. [2024], Zheng et al. [2024]. Note also that Assumption 1 does not impose structural constraints such as simultaneous diagonalizability on the task matrices.

### 2.2 Linear transformer architecture

We first recall the construction of a linear transformer block. Let $(e_1, \ldots, e_t)$ be a sequence of tokens. A linear transformer block with attention weights $W_P, W_Q, W_V, W_K$ and MLP weights $W_1, W_2$ maps the sequence $(e_1, \ldots, e_t)$ to the sequence $(\widehat{e}_1, \ldots, \widehat{e}_T)$ given by

$$\widehat{e}_t = W_{MLP} \left( e_t + W_P W_V E_t \cdot \frac{E_t^T W_K^T W_Q e_t}{\rho_t} \right),$$

where $E_t = (e_1, \ldots, e_t)$. Note that each output token $\widehat{e}_t$ is a function of only the first $t$ input tokens. For the purpose of our analysis, we simplify the structure of the attention layer by re-parameterizing some of the weight matrices. The simplified linear transformer block has attention weights $W_P, W_Q$ and MLP weights $W_{MLP}$ and is given by

$$\widehat{e}_t = W_{MLP} \left( e_t + W_P E_t \cdot \frac{1}{\rho_t} E_t^T W_Q e_t \right).$$

Note that from an approximation theory point of view, this re-parameterization of the weights does not lose us any expressiveness. In the case of learning the dynamical system defined by Equation (2), the observations $(x_0, x_1, \ldots, x_T)$ are encoded into a sequence of tokens $(e_1, \ldots, e_T)$, which the user has the freedom to specify. We consider the following positional encoding given by $e_t = (0_d^T, 0_d^T, x_t^T, x_{t-1}^T)^T \in \mathbb{R}^{4d}$. The first $d$ rows of the token $e_t$ serve as a placeholder for the transformer's prediction for the next token. That is, if $\widehat{e}_t$ is the $t^{\text{th}}$ element of the sequence defined by the transformer block, we write $\widehat{y}_t = (\widehat{e}_t)_{1:d}$ for the prediction of $x_{t+1}$. We note that a similar prompt embedding for autoregressive transformers, in which only the first $d$ rows of $e_t$ are held as zero, has been studied in several works Von Oswald et al. [2023b], Sander et al. [2024], Zheng et al. [2024]. The reason for choosing this specific embedding structure (with zeros in the first $2d$ entries) will become clear in the proof of Theorem 1. We take the normalization factor to be $\rho_t = t$.

A multi-layer linear transformer is then defined simply by composing linear transformer blocks. More precisely, an $L$-layer transformer is parameterized by weights

$$\{W_{MLP}^{(\ell)}, W_P^{(\ell)}, W_Q^{(\ell)} \in \mathbb{R}^{4d \times 4d} : 1 \leq \ell \leq L\}.$$

As in the single-layer case, we use the first $d$ rows of the Transformer output of each token $\widehat{y}_t = (\widehat{e}_t)_{1:d}$ for prediction. Denote by $\mathrm{TF}_L$ the class of $L$-layer linear transformers.

## 2.3 Training and inference loss functions

Transformers are typically trained to perform in-context learning by minimizing the following *auto-regressive training loss*

$$L_{\mathrm{train}}(\theta) = \mathbb{E}_{(\xi_1,\ldots,\xi_T,W)} \left[ \frac{1}{T-1} \sum_{t=1}^{T-1} \|\widehat{y}_t - x_{t+1}\|^2 \right], \tag{3}$$

where the transformer learns to predict the next token $x_{t+1}$ from the previous $t$ tokens. Here, $\theta$ denotes the transformer parameters. In the setting where $(x_1, \ldots, x_T)$ follow a dynamical system of the form in Equation (2), we emphasize that the transformer is learning to predict the mean of $x_{t+1}$ conditioned on $x_t$ for each $t \in \{1, \ldots, T-1\}$. This follows from the fact that the loss $L(\theta)$ averages over the measurement noise. To analyze the in-context learning ability of a given transformer, we define the $L^2$ *test loss*

$$L_T(\theta) = \sup_{W \in \mathcal{W}} \mathbb{E}_{\xi_1,\ldots,\xi_t} \left[ \|\widehat{y}_T - W x_T\|^2 \right]. \tag{4}$$

The test loss admits three key differences from the train loss. First, it measures the difference between the transformer prediction $\widehat{y}_t$ and the conditional expectation $W x_t = \mathbb{E}[x_{t+1}|x_t, W]$, rather than the difference between $\widehat{y}_t$ and the noisy label $x_{t+1}$ used in the training loss. By a simple calculation, this is equivalent to defining the loss using the difference between $\widehat{y}_t$ and $x_{t+1}$, up to an additive factor which depends only on the distribution of the noise. Thus, this change can be viewed as a centering of the loss to ensure that the minimum of the test loss tends to zero as $T \to \infty$. Second, the test loss only measures the error of the transformer prediction at the terminal time $T$. Since the transformer prediction $\widehat{y}_t$ is only a function of the first $t$ iterates of the dynamical system $(x_1, \ldots, x_t)$, we cannot expect the transformer to make accurate predictions when $t$ is small. Third, and perhaps most importantly, the test loss measures the error *uniformly* over the task set $\mathcal{W}$. This is quite different from the typical $L^2$-loss used to measure the prediction error in supervised learning. However, a small $L^2$-loss does not provide guarantees on the model's inference-time prediction on a given downstream task $W \in \mathcal{W}$. In contrast, the uniform loss captures the worst-case robustness of the model, similar to the minimax paradigm in nonparametric statistics Giné and Nickl [2021]. Under the interpretation of transformers as algorithms, we therefore view the uniform-in-$W$ loss as a natural metric for in-context learning. We note that while the uniform loss is difficult to use for training purposes, several works have proven that transformers trained on an $L^2$-loss do in fact exhibit inference-time guarantees which hold uniformly over the task space Zhang et al. [2023], Yang et al. [2024], Li et al. [2024].

## 3 Main results

### 3.1 Approximation error bound for deep linear transformers

Our first theoretical result is an upper bound of the approximation error of deep transformers in learning the dynamical system (2) in-context, measured by the test loss (4).

**Theorem 1.** *There exists a transformer of depth $L = O(\log(T))$ with parameters $\theta$ such that*

$$L_T(\theta) \lesssim \frac{\log(T)}{T}.$$

*when $T$ is sufficiently large. The implicit constants depend on $\sigma$, $w_{\mathrm{max}}$, and $d$, and the dependence is at most polynomial in $d$.*

Theorem 1 shows that there is a transformer that can closely track the condition mean as $T$ increases to infinity. The key idea underlying the proof of Theorem 1 is to construct a transformer such that, for any dynamical system $(x_1, \ldots, x_T)$ according to (2), the transformer prediction $\widehat{y}_t$ approximates

the least-squares prediction $\widehat{W} x_t$, where $\widehat{W} = \operatorname{argmin}_W \frac{1}{t} \sum_{i=0}^{t-1} \| x_{i+1} - W x_i \|^2$ is the least-squares estimator. The proof then proceeds by leveraging the statistical properties of the least-squares estimator to achieve the final bound. Implementing the least squares estimator requires computing the matrix vector product $X_t^{-1} x_t$, where $X_t = \frac{1}{t} \sum_{i=0}^{t-1} x_i x_i^T$ is the empirical covariance matrix. An important ingredient of the proof of Theorem 1 is the construction of a transformer which unrolls a *modified Richardson iteration* algorithm for solving the linear system $X_t b = x_t$ for $X_t^{-1} x_t$. The convergent guarantees of the Richardson iteration allow us to derive quantitative bounds between the transformer prediction and the least-squares estimator. We note that the algorithm unrolling idea has been used to prove the approximation power of deep neural networks Chen et al. [2021], Marwah et al. [2023] for solutions of PDEs. The recent works adopted the unrolling idea for proving the approximation power of transformers for nonlinear regression tasks Bai et al. [2024]. The recent paper Von Oswald et al. [2023b] also utilized unrolling of the Richardson's iteration in their construction of a transformer for learning deterministic linear dynamical systems, but did not provide an approximation guarantee for the constructed transformer.

The error achieved by the deep transformer constructed in Theorem 1 decays at the parametric rate $O(1/T)$, up to a logarithmic factor. This is not surprising, as our proof strategy constructs a transformer which (approximately) implements a parametric estimator. Theorem 1 also shows that the depth of the transformer needed to achieve this rate is only logarithmic in the sequence length $T$, owed to the fast convergence of the Richardson iteration. Interestingly, the recent work Yüksel et al. [2024] studies statistical guarantees of the least-squares predictor for long-context problems, and their results could be potentially leveraged to generalize our approximation results to long-context systems. We leave this extension to future work.

## 3.2 A lower bound for single-layer transformers

While it is beyond the scope of this paper to characterize whether $O(\log(T))$ depth is necessary for in-context learning dynamical systems, we provide an analysis for the (limited) approximation power of single-layer transformers. To be concrete, we focus on in-context learning of the one-dimensional dynamical system

$$\begin{cases} x_t = w x_{t-1} + \xi_t, \ t \in \{1, \ldots, T\}, \\ x_0 = 0, \end{cases} \tag{5}$$

where $\xi_t \sim N(0, \sigma^2)$ and $w \in [w_{\min}, w_{\max}]$. To make our analysis tractable, we fix some of the parameters of the single-layer transformer; afterwards, we discuss how our results might be extended to the general case. First, we fix the MLP weight matrix $W_{MLP}$ to be the $d \times d$ identity matrix. The resulting architecture is often referred to as a *linear attention block* and has been the study of a lot of recent research Zhang et al. [2023], Mahankali et al. [2023], Zheng et al. [2024], Zhang et al. [2024]. In the 1D setting, a linear attention block defines an estimator $\widehat{y}_T$ for $x_{T+1}$ given by

$$\widehat{y}_t = \frac{1}{T} \begin{pmatrix} p_1 & p_2 \end{pmatrix} \cdot \begin{pmatrix} \sum_{i=1}^T x_i^2 & \sum_{i=1}^T x_i x_{i-1} \\ \sum_{i=1}^T x_i x_{i-1} & \sum_{i=1}^T x_{i-1}^2 \end{pmatrix} \cdot \begin{pmatrix} q_{11} & q_{12} \\ q_{21} & q_{22} \end{pmatrix} \cdot \begin{pmatrix} x_T \\ x_{T-1} \end{pmatrix},$$

where $p_i \in \mathbb{R}$ and $q_{ij} \in \mathbb{R}$ are the learnable parameters. Since we know that $\mathbb{E}[x_{t+1} | x_0, x_1, \ldots, x_t]$ is a linear function of $x_t$, it is natural to zero out the parameters that lead to $\widehat{y}_t$ being a linear combination of both $x_t$ and $x_{t-1}$. This motivates us to further set $q_{12} = q_{22} = 0$, leading to the simplified parameterization

$$\widehat{y}_T = \frac{1}{T} \mathbf{p}^T \begin{pmatrix} \sum_{i=1}^T x_i^2 & \sum_{i=1}^T x_i x_{i-1} \\ \sum_{i=1}^T x_i x_{i-1} & \sum_{i=1}^T x_{i-1}^2 \end{pmatrix} \mathbf{q} \cdot x_T, \tag{6}$$

where $\mathbf{p} = \begin{pmatrix} p_1 & p_2 \end{pmatrix}^T$ and $\mathbf{q} = \begin{pmatrix} q_1 & q_2 \end{pmatrix}^T$. We note that a similar parametrization was also adopted in Zheng et al. [2024] and it was proven that a single layer of fully-parameterized linear attention trained on gradient flow of the $L^2$ loss will converge to a parameterization where only $p_{21}$ and $q_{11}$ are nonzero. In the one-dimensional setting, the test loss introduced in Subsection 2.3 becomes

$$L_T(\mathbf{p}, \mathbf{q}) = (\widehat{y}_T - w x_T)^2.$$

Our aim is to understand how the approximation error $\inf_{\mathbf{p}, \mathbf{q}} L_T(\mathbf{p}, \mathbf{q})$ behaves as a function of $T$. Specifically, we shall show that $\inf_{\mathbf{p}, \mathbf{q}} L_T(\mathbf{p}, \mathbf{q}) = \Omega(1)$ as $T \to \infty$, from which we conclude that the single-layer transformer induces an irreducible approximation error, indicating that it is

insufficient in learning the linear dynamical system. To this end, we first express the test loss function as $L_T(\mathbf{p}, \mathbf{q}) = \mathbb{E}[\ell(\mathbf{p}, \mathbf{q}, w)]$, where the individual loss is defined as

$$\ell_T(\mathbf{p}, \mathbf{q}, w) = (\widehat{y}_T - wx_T)^2.$$

The following lemma characterizes the limiting behavior of the individual loss as $T \to \infty$.

**Proposition 1.** *The individual loss function converges pointwise on $\mathbb{R}^2 \times \mathbb{R}^2 \times [w_{\min}, w_{\max}]$ to a limiting function $\ell(\mathbf{p}, \mathbf{q}, w)$ defined by*

$$\ell(\mathbf{p}, \mathbf{q}, w) = \frac{\sigma^2}{1 - w^2} \left( \frac{\sigma^2}{1 - w^2} (w\alpha_1 + \alpha_2) - w \right)^2,$$

*where $\alpha_1 = p_1 q_2 + p_2 q_1$ and $\alpha_2 = \mathbf{p}^T \mathbf{q}$. Moreover, $\ell$ satisfies the lower bound*

$$\inf_{\mathbf{p}, \mathbf{q} \in \mathbb{R}^2} \sup_{w \in [w_{\min}, w_{\max}]} \ell(\mathbf{p}, \mathbf{q}, w) \geq C(\sigma^2, w_{\min}, w_{\max}),$$

*where $C(\sigma^2, w_{\min}, w_{\max}) > 0$ is a strictly positive constant depending only on $\sigma^2$, $w_{\min}$, and $w_{\max}$.*

The proof of the formula for $\ell(\mathbf{p}, \mathbf{q}, w)$ follows from a careful computation of the expectations defining the test loss. The upshot of Proposition 1 is that it allows us to study the behavior of the minimum value of the test loss $L_T$ as the sequence length tends to infinity. In particular, we can translate the lower bound in Proposition 1 into a lower bound on the test loss in the limit $T \to \infty$. This is the content of our main theoretical result in this section, which we state below.

**Theorem 2.** *For any $R > 0$, the limit $\lim_{T \to \infty} \inf_{\|\mathbf{p}\|, \|\mathbf{q}\| \leq R} L_T(\mathbf{p}, \mathbf{q})$ exists, and*

$$\lim_{T \to \infty} \inf_{\|\mathbf{p}\|, \|\mathbf{q}\| \leq R} L_T(\mathbf{p}, \mathbf{q}) \geq C(\sigma^2, w_{\min}, w_{\max}),$$

*where $C(\sigma^2, w_{\min}, w_{\max}) > 0$ is the constant from Proposition 1.*

For technical reasons, we must constrain attention parameters $\mathbf{p}, \mathbf{q}$ to lie in a ball of radius $R$ for Theorem 2. It is common in deep learning that model parameters are constrained either by explicit or implicit regularization. Note, however, that the constant in the lower bound of Theorem 1 is independent of $R$, so $R$ can be arbitrarily large.

Theorem 2 shows that a single layer of linear attention is unable to accurately capture the dynamics defined by the system (5). To the best of our knowledge, Theorem 2 provides the first *lower bound* for in-context learning of dynamical systems, and it suggests that some care is required in choosing an appropriate architecture in this setting. While our results in this section are stated for 1D dynamical systems, we expect that the results generalize to arbitrary dimensions. For illustrative purposes, let us consider a $d$-dimensional dynamical system $(x_1, \ldots, x_T)$ according to Equation (2) where the task matrix $W \in \mathbb{R}^{d \times d}$ is diagonal. Then, since the noise is isotropic, this is equivalent to specifying a system of $d$ independent 1D dynamical systems of the form in Equation (5). We therefore expect that in-context learning a multi-dimensional dynamical system is at least as difficult as in-context learning a system of 1D dynamical, for which we have lower bounds. We leave it to future work to rigorously generalize Theorem 2 to higher dimensions.

In addition, we expect that the non-zero lower bound on the limiting loss can be extended to general single layer parameters (i.e., fully parameterized attention with an MLP matrix). At a high level, the result in Proposition 1 follows from the fact that $\ell(\mathbf{p}, \mathbf{q}, w)$ is a polynomial in the parameters, but a non-polynomial function in $w$ (in particular, it depends on $(1 - w^2)^{-1}$). Since this is still the case if we consider a fully-parameterized single-layer linear transformer with an MLP matrix, we expect an analogous result to hold for general single-layer linear transformers. However, the explicit computation of the limiting loss is more complicated, and we leave it as an interesting avenue for future work to justify this rigorously.

### 3.3 Comparing in-context learning over IID vs non-IID data:

Our theoretical results hold for non-independent data generated according to a dynamical system. A more common setting for theoretical analysis is the in-context learning of IID data, where a model, given a prompt of independent observations $((x_1, f(x_1), \ldots, (x_T, f(x_T)))$ of some unseen task $f$

and an unlabeled query point $x_{T+1}$, is asked to predict the new label $f(x_{T+1})$. Several recent works Zhang et al. [2023], Ahn et al. [2023], Mahankali et al. [2023] have analyzed this problem when the task $f$ is given by a linear map $f(x) = \langle w, x \rangle$. In this case, it has been shown that a single layer of linear attention can achieve 0 test error in the limit $T \to \infty$, even when the loss is measured by the uniform norm on the task space. In contrast, we have shown in Theorem 2, for an analogous problem involving non-IID data, that a single layer of linear attention can never achieve zero test loss in the large sample limit. The intuitive explanation for such a discrepancy can be understood as follows. In the IID setting, it has been shown that the optimal parameterization of a linear attention layer depends on the distribution $P_X$ of the covariates $\{x_1, x_2, \ldots, x_T\}$. Specifically, Zhang et al. [2023] shows that a block of the optimal attention matrix $W_Q$ learns to invert the covariance matrix $\text{Cov}(P_X)$, which is independent of the task vector $w \in \mathbb{R}^d$. For dynamical systems $(x_1, \ldots, x_T)$ given by Equation (2), this construction fails to perform in-context learning because *the covariate distribution depends on the task matrix $W$*. This indicates that transformers may have more difficulty performing in-context learning on non-IID data compared to IID data.

# 4 Numerical experiments

In this section, we present numerical experiments to validate the claims of Theorems 1 and 2. We train transformers of varying depth to in-context learn noisy one-dimensional linear dynamical systems of the form (2), and we plot the test error of the trained transformers (as measured by the $L^2$ norm over both $\{x_t\}$ and the task $w$) with $T = 500$ and $[w_{\min}, w_{\max}] = [0, 0.8]$ as a function of the number of epochs. The results are displayed in Figure 1. Both softmax and linear transformers exhibit a clear separation between their optimal test loss at depth $L = 1$ and $L \geq 2$, which supports Theorem 2. For softmax transformers, the optimal test loss continues to decrease as a function of depth up to $L = 5$. For linear transformers, the optimal loss is approximately the same at $L = 2, 3, 4, 5$, and it is difficult to say whether this is due to the capacity of deep linear transformers, or simply a manifestation of optimization error. Our experiments with linear transformers suggest that our result in Theorem 1 may be sup-optimal, and that there may exist a transformer with constant depth which achieves vanishing test loss. On the other hand, our results on softmax transformers suggest that increasing depth beyond $L = 2$ improves the test error; this is in contrast to empirical findings for discrete Markov chains Olsson et al. [2022], where an induction head emerges at depth $L = 2$.

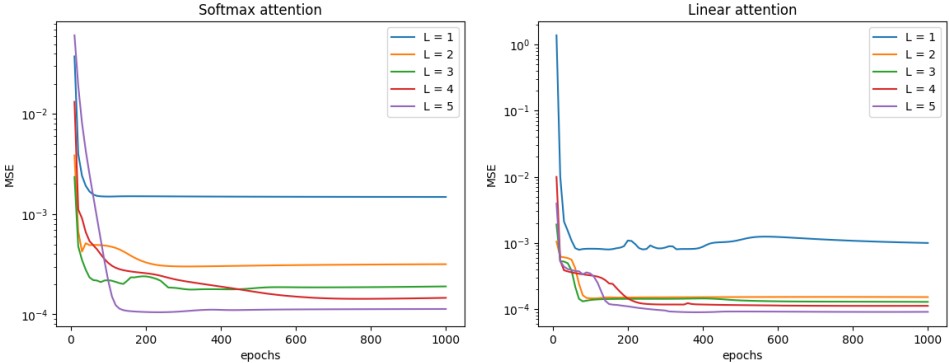

Figure 1: Test error as a function of training epochs, with $d = 1, T = 500, [w_{\min}, w_{\max}] = [0, 0.8]$, and various values of $L$

# 5 Proof sketches

## 5.1 Proof sketch for Theorem 1

As noted in Section 3.1, our proof of Theorem 1 leverages *algorithm unrolling*, a powerful technique for proving approximation error bounds whereby a deep model encodes a learning algorithm, with each layer of the model corresponding to an iteration of the algorithm. Specifically, we use a transformer to unroll the least-squares algorithm, which approximates the matrix $W$ defining the

dynamical system define in Equation (2) given the history up to time $T$ by

$$\widehat{W}_T := \text{argmin}_W \frac{1}{T} \sum_{i=0}^{T-1} \|x_{i+1} - W x_i\|^2 = \left(\frac{1}{T} \sum_{i=0}^{T-1} x_{i+1} x_i^T\right) \left(\frac{1}{T} \sum_{i=0}^{T-1} x_i x_i^T\right)^{-1}.$$

Given the history of the dynamical system $(x_0, \ldots, x_T)$ up to time $T$, the goal of the transformer is to predict the conditional expectation $\mathbb{E}[x_{T+1}|x_T] = W x_T$. In light of the discussion above natural estimator for $x_{T+1}$ is $\widehat{W}_T X_T$. Let $X_T = \frac{1}{T} \sum_{i=0}^{T-1} x_i x_i^T$ denote the sample covariance matrix of the dynamical system, and assume for the moment that $X_T$ is invertible. Note that computing $\widehat{W}_T x_T$ requires one to find a vector $z_T$ such that $X_T^{-1} x_T = z_T$, or $X_T z_T = x_T$. A classical iterative method for solving linear systems of this form is the *modified Richardson iteration*. Given a linear system $Ax = b$, the modified Richardson iteration with step size $\alpha > 0$ approximates the solution $x$ by the update rule $x_{(\ell)} = (\mathbf{I}_d - \alpha A)x_{(\ell-1)} + \alpha b$. A key technical lemma in proving Theorem 1 constructs a transformer which approximates the least-squares prediction $\widehat{W}_T x_T$ by unrolling the modified Richardson iteration. Let us denote by $z_{T,L}$ the $L^{\text{th}}$ step of the modified Richardson iteration

$$\begin{cases} z_{T,\ell} = (\mathbf{I}_d - \alpha X_T) z_{T,\ell-1} + \alpha x_T, \ \ell \geq 0 \\ z_{T,0} = 0. \end{cases}$$

We state the result precisely below.

**Lemma 1.** *For any positive integer $L$ and $\alpha > 0$, there exists an $(L+1)$-layer transformer with parameters $\theta^* = \{W_{MLP}^{(\ell)}, W_P^{(\ell)}, W_Q^{(\ell)}\}_{\ell=1}^{L+1}$ such that for any dynamical system $(x_0, x_1, \ldots, x_T)$ defined by (2), $\theta^*$ maps the sequence $(e_1, \ldots, e_T)$ defined by Equation (2) to a sequence $(\widehat{e}_1, \ldots, \widehat{e}_T)$ where, for each $1 \leq t \leq T$*

$$\widehat{y}_t := (\widehat{e}_t)_{1:d} = \left(\frac{1}{t} \sum_{i=0}^{t-1} x_{i+1} x_i^T\right) z_{t,L}.$$

*In other words, there exists a transformer which estimates $x_{T+1}$ by 1) approximating the matrix-vector product $X_T^{-1} x_T$ via the Richardson iteration, and 2) using the approximation to $X_T^{-1} x_T$ to compute an approximation to the least-squares prediction $\widehat{x}_{T+1}$ for $x_{T+1}$.*

To translate Lemma 1 into a bound on the test loss $L_T(\theta)$, there are several technical hurdles. First, in order for the least-squares estimator to be well-defined, the sample covariance matrix $x_T$ must be invertible. In this case, if we hope to obtain any quantitative bounds, we need upper bounds on the condition number of the sample covariance (which governs the rate of convergence of the modified Richardson iteration) and estimates on the statistical performance of the least-squares estimator. To this end, we apply results from Foster et al. [2020] and Matni and Tu [2019], which provide high-probability guarantees on the condition number of $X_T$ and the covariance and the discrepancy $\|\widehat{W}_T - W\|$, respectively. If we let $\mathcal{A}$ denote the event on which these hold (see a precise definition in Appendix A), then, for fixed $W \in \mathcal{W}$, we can bound the $L^2$-error of the transformer by

$$\mathbb{E}\left[\|\widehat{y}_t - W x_T\|^2\right] \leq 2\mathbb{E}\left[\|\widehat{y}_t - W x_T\|^2 \cdot 1_{\mathcal{A}}\right] + 2\mathbb{E}\left[\|(\widehat{y}_t - W x_T)\|^2 \cdot 1_{\mathcal{A}^c}\right].$$

On the event $\mathcal{A}$, we can bound the expectation further by

$$\mathbb{E}\left[\|\widehat{y}_t - W x_T\|^2 \cdot 1_{\mathcal{A}}\right] \leq 2\mathbb{E}\left[\left\|\widehat{y}_t - \widehat{W}_T x_T\right\|^2 \cdot 1_{\mathcal{A}}\right] + 2\mathbb{E}\left[\left\|\left(\widehat{W}_T - W\right) x_T\right\|^2 \cdot 1_{\mathcal{A}}\right].$$

The first term above can be bounded by leveraging the convergence rate of the modified Richardson iteration, while the second term can be bounded by applying the bound on $\|\widehat{W}_T - W\|$. The two terms can be balanced by choosing the step size as an appropriate function of $T$. When $L = O(\log(T))$, this yields a bound of $O\left(\frac{\log(T)}{T}\right)$. To bound the expectation on $\mathcal{A}$, we can apply the Cauchy-Schwartz inequality

$$\mathbb{E}\left[\|(\widehat{y}_t - W x_T)\|^2 \cdot 1_{\mathcal{A}^c}\right] \leq \mathbb{E}\left[\|(\widehat{y}_t - W x_T)\|^4\right]^{1/2} \cdot \mathbb{P}\left(\mathcal{A}^c\right)^{1/2}.$$

There is some care required to handle this term, because, away from the event $\mathcal{A}$ the norm of $\widehat{y}_T$ can grow with the step size $L$, which could potentially offset the decay of $\mathbb{P}\left(\mathcal{A}^c\right)^{1/2}$. In Lemma 11 in

Appendix C, we bound the moments of the Richardson iterate $z_{T,L}$, which allows us to prove that the above term is $O\left(\frac{\log(T)}{T}\right)$ when $L = O(\log(T))$. This proves that $\mathbb{E}\left[\|\widehat{y}_t - W x_T\|^2\right] = O\left(\frac{\log(T)}{T}\right)$, for each $W \in \mathcal{W}$, and taking the supremum over $\mathcal{W}$ gives the result of Theorem 1. See Appendix A for the detailed proof.

## 5.2 Proof sketch for Theorem 2

A great deal of the technical work in proving Theorem 2 lies in computing the limit of the individual loss function in Lemma 1:

$$\lim_{T \to \infty} \ell_T(\mathbf{p}, \mathbf{q}, w) := \ell(\mathbf{p}, \mathbf{q}, w) = \frac{\sigma^2}{1 - w^2}\left(\frac{\sigma^2}{1 - w^2}(w\alpha_1 + \alpha_2) - w\right)^2,$$

where $\alpha_1 = p_1 q_2 + p_2 q_1$ and $\alpha_2 = \mathbf{p}^T \mathbf{q}$. Since these computations are cumbersome, we defer them to the Appendix (see Appendix B and also Appendix C for proofs of auxiliary moment computations). While the closed form expression for $\ell_T(\mathbf{p} < \mathbf{q}, w)$ is a lengthy sum of many different terms, the ergodicity of the dynamical system ensures that only a few terms survive in the limit; this is an important theme of our computations. Once this formula is established, the lower bound on $\inf_{\mathbf{p},\mathbf{q}} \sup_{w \in [w_{\min}, w_{\max}]} \ell(\mathbf{p}, \mathbf{q}, w)$ stated in Lemma 1 can be argued as follows: for any finite collection $\{w_1, \ldots, w_K\} \in [w_{\min}, w_{\max}]$, we have the lower bound

$$\inf_{\mathbf{p},\mathbf{q}} \sup_{w \in [w_{\min}, w_{\max}]} \ell(\mathbf{p}, \mathbf{q}, w) \geq \inf_{\alpha_1, \alpha_2 \in \mathbb{R}} \max_{1 \leq i \leq K} \frac{\sigma^2}{1 - w_i^2}\left(\frac{\sigma^2}{1 - w_i^2}(w_i \alpha_1 + \alpha_2) - w_i\right)^2.$$

In other words, we replace the infimum over $\mathbf{p}, \mathbf{q} \in \mathbb{R}^2$ with the infimum over all $\alpha_1, \alpha_2 \in \mathbb{R}$ and we replace the supremum over $[w_{\min}, w_{\max}]$ with a finite maximum. The infimum is attained at a pair $(\alpha_1^*, \alpha_2^*)$ at which the graphs of a subset of the $K$ curves intersects (see Equation (20) in Appendix B for details here). For appropriately chosen $w_1, \ldots, w_K$, it can be shown that this infimum is equal to zero if and only if $w_1, \ldots, w_K$ solve a certain linear system. The lower bound then follows from recognizing that the system is inconsistent. Once the lower bound in Lemma 1 is proven, Theorem 2 can be proven by using the regularity of the family of individual loss functions $\{(\mathbf{p}, \mathbf{q}, w) \mapsto \ell_T(\mathbf{p}, \mathbf{q}, w)\}$ to interchange limits and suprema/infima.

## 6 Conclusion and discussion

We studied the approximation power of transformers for performing in-context learning on data arising from linear dynamical systems. For multilayer transformers, we showed that logarithmic depth is sufficient to achieve fast decay of the test loss as the context length tends to infinity. Conversely, we proved a lower bound for single-layer linear transformers which suggest their incapability of learning such dynamical systems in-context. We also provided numerical results that confirmed the benefits of increasing the depth of the transformer in improving the prediction performance. There are several important directions for future research. First, we would like to better understand the apparent depth-separation observed in this paper. In particular, it remains to be determined whether there is a transformer with $O(1)$ depth whose test loss vanishes as the sequence $T \to \infty$. If this question can be answered affirmatively, we would also like to describe the mesa-optimization algorithm that such a transformer encodes. Second, it would be interesting to generalize the analysis of this paper to *nonlinear* dynamical systems. We anticipate the the unrolling idea may still be effective, but carrying this out is highly non-trivial as the least-square estimator does not admit a closed form in the nonlinear setting. Finally, although this paper focused on the approximation only, it remains an open question to investigate whether the transformers trained on in-context examples of the linear dynamical system (2) can in-context learn the dynamical system, in the sense of the uniform loss in Equation (4). We leave these various problems to future work.

## 7 Acknowledgment

YL thanks the support from the NSF CAREER Award DMS-2442463 and the support from the Data Science Initiative at University of Minnesota through a MnDRIVE DSI Seed Grant. The authors thank Shaurya Sehgal for sharing his codes used to produce preliminary numerical results.

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

# A    Proofs for Section 3.1

We recall some notation from Section 5.1. We let $X_T = \frac{1}{T} \sum_{i=0}^{T-1} x_i x_i^T$ denote the sample covariance matrix of the dynamical system and $\widehat{W}_T$ denote the least-squares estimator of the dynamical system given by

$$\widehat{W}_t = \text{argmin}_W \sum_{t=0}^{T-1} \|x_{i+1} - W x_i\|^2 = \left( \frac{1}{T} \sum_{i=0}^{T-1} x_{i+1} x_i^T \right) \left( \underbrace{\frac{1}{T} \sum_{i=0}^{T-1} x_i x_i^T}_{=X_T} \right)^{-1}. \tag{7}$$

We let $z_{T,L}$ be defined by the modified Richardson iteration

$$\begin{cases} z_{T,L} = z_{T,L-1} + \alpha \left( x_T - X_T z_{T,L-1} \right), \ L > 0 \\ z_{T,0} = 0. \end{cases} \tag{8}$$

We make use of the following classical result on the convergence of the modified Richardson iteration.

**Lemma 2.** *[Richardson iteration, Varga [1962]] Assume that the matrix $A \in \mathbb{R}^{d \times d}$ is invertible let $\alpha > 0$ be small enough that so that $\|\mathbf{I}_d - \alpha A\|_{op} < 1$. Let $x_* \in \mathbb{R}^d$ denote the solution to the equation $Ax = b$ and let $x_{(k)}$ denote the vector obtained at the $k^{th}$ step of Richardson iteration with initialization $x_0 = 0$. Then $x_k$ takes the explicit form*

$$x_{(k)} = \left( \mathbf{I}_d - \left( \mathbf{I}_d - \alpha A \right)^k \right) x_*,$$

*and consequently, the error bound*

$$\|x_* - x_{(k)}\| \leq \|\mathbf{I}_d - \alpha A\|_{op}^k \|x_*\|$$

*holds for all $k$.*

In particular, Lemma 2 shows that the modified Richardson iterates converge to the true solution of the linear system at an exponential rate. For a fixed matrix $A$, the optimal step size $\alpha$ depends on the smallest and largest eigenvalues of $A$.

We are interested in constructing a transformer which approximately implements the least-squares prediction $\widehat{W}_T x_T$ to estimate $x_{T+1}$. In order to approximate the term $X_T^{-1} x_T$ appearing in the expression for $\widehat{W}_T x_T$, we design a transformer which unrolls the modified Richardson iteration to solve a related linear system. Here, depth plays a crucial role, as each layer of the transformer represents a step of the modified Richardson iteration. Our result is stated precisely in Lemma 1 in Section 5.1. We now present the proof of this result.

*Proof of Lemma 1.* **Step 1:** Let $a_1, \ldots, a_T, b_1, \ldots, b_T \in \mathbb{R}^d$ denote fixed vectors. We first show that for any sequence of tokens $(e_1, \ldots, e_t)$ where $e_i = (a_t, b_t, x_t, x_{t-1})^T$, there exists a single layer transformer which maps the sequence $(e_1, \ldots, e_T)$ to $(\widehat{e}_1, \ldots, \widehat{e}_T)$, where

$$\widehat{e}_t = \begin{pmatrix} a_t + \alpha(x_t - \alpha X_t a_t) \\ b_t + \alpha(x_t - X_t b_t) \\ x_t \\ x_{t-1} \end{pmatrix}.$$

In words, there exists a single-layer transformer block which simultaneously implements one step of the Richardson iteration for $X_{t-1}^{-1} x_{t-1} = b$ initialized at $a_t$ and $b_t$ with step size $\alpha$. To prove this, let us first define the attention weights by

$$W_P = \begin{pmatrix} 0 & 0 & 0 & \mathbf{I}_d \\ 0 & 0 & 0 & \mathbf{I}_d \\ 0 & 0 & 0 & 0 \\ 0 & 0 & 0 & 0 \end{pmatrix} \in \mathbb{R}^{4d \times 4d}, \quad W_Q = \begin{pmatrix} 0 & 0 & 0 & 0 \\ 0 & 0 & 0 & 0 \\ 0 & 0 & 0 & 0 \\ -\alpha \mathbf{I}_d & 0 & 0 & 0 \end{pmatrix} \in \mathbb{R}^{4d \times 4d}.$$

Then we have

$$W_P \cdot E_t E_t^T \cdot W_Q e_t = \begin{pmatrix} 0 & 0 & 0 & \mathbf{I}_d \\ 0 & 0 & 0 & \mathbf{I}_d \\ 0 & 0 & 0 & 0 \\ 0 & 0 & 0 & 0 \end{pmatrix} \cdot \frac{1}{t} \sum_{i=1}^{t} \begin{pmatrix} a_i \\ b_i \\ x_i \\ x_{i-1} \end{pmatrix} \begin{pmatrix} a_i^T & b_i^T & x_i^T & x_{i-1}^T \end{pmatrix} \cdot \begin{pmatrix} 0 & 0 & 0 & 0 \\ 0 & 0 & 0 & 0 \\ 0 & 0 & 0 & 0 \\ -\alpha \mathbf{I}_d & 0 & 0 & 0 \end{pmatrix} \begin{pmatrix} a_t \\ b_t \\ x_t \ x_{t-1} \end{pmatrix}$$

$$= \frac{1}{t} \sum_{i=1}^{t} \begin{pmatrix} x_{i-1} \\ x_{i-1} \\ 0 \\ 0 \end{pmatrix} \begin{pmatrix} -\alpha x_{i-1}^T & 0^T & 0^T & 0^T \end{pmatrix} \begin{pmatrix} a_t \\ b_t \\ x_t \\ x_{t-1} \end{pmatrix}$$

$$= \frac{1}{t} \sum_{i=1}^{t} \begin{pmatrix} -\alpha x_{i-1} x_{i-1}^T & 0 & 0 & 0 \\ -\alpha x_{i-1} x_{i-1}^T & 0 & 0 & 0 \\ 0 & 0 & 0 & 0 \\ 0 & 0 & 0 & 0 \end{pmatrix} \begin{pmatrix} a_t \\ b_t \\ x_t \\ x_{t-1} \end{pmatrix}$$

$$= \begin{pmatrix} -\alpha X_t a_t \\ -\alpha X_t b_t \\ 0 \\ 0 \end{pmatrix}.$$

It follows that

$$e_t + W_P \cdot E_t E_t^T \cdot W_Q e_t = \begin{pmatrix} a_t - \alpha X_t a_t \\ b_t - \alpha X_t b_t \\ x_t \\ x_{t-1} \end{pmatrix}.$$

Now, let $W_{MLP} \in \mathbb{R}^{4d \times 4d}$ be any matrix such that

$$W_{MLP} \begin{pmatrix} a_t - \alpha X_t a_t \\ b_t - \alpha X_t b_t \\ x_t \\ x_{t-1} \end{pmatrix} = \begin{pmatrix} a_t + \alpha(x_t - X_t a_t) \\ b_t + \alpha(x_t - X_t b_t) \\ x_t \\ x_{t-1} \end{pmatrix}.$$

Then the transformer block with weights $W_{MLP}, W_P, W_Q$ maps $e_t$ to

$$\widehat{e}_t = W_{MLP} \left( e_t + W_P \cdot E_t E_t^T \cdot W_Q e_t \right)$$

$$= W_{MLP} \begin{pmatrix} a_t - \alpha X_t a_t \\ b_t - \alpha X_t b_t \\ x_t \\ x_{t-1} \end{pmatrix}$$

$$= \begin{pmatrix} a_t + \alpha(x_t - X_t a_t) \\ b_t + \alpha(x_t - X_t b_t) \\ x_t \\ x_{t-1} \end{pmatrix}.$$

**Step 2:** By composing the transformer blocks from Step 1, we have shown that there exists an $L$-layer transformer which maps the token $e_t = \begin{pmatrix} 0 \\ 0 \\ x_t \\ x_{t-1} \end{pmatrix}$ to the token

$$\widehat{e}_t^{(L)} = \begin{pmatrix} z_{t,L} \\ z_{t,L} \\ x_t \\ x_{t-1}, \end{pmatrix}$$

where $z_{t,L}$ Richardson iterate defined in Equation (8).

**Step 3:** We now show that for any sequence of tokens $(e_1, \ldots, e_T)$ with $e_i = (a_i, a_i, x_i, x_{i-1})^T$, there exists a single layer transformer block which maps $(e_1, \ldots, e_T)$ to $(\widehat{e}_1, \ldots, \widehat{e}_T)$, where

$$(\widehat{e}_t)_{1:d} = \left( \frac{1}{t} \sum_{i=0}^{t-1} x_{i+1} x_i^T \right) a_t.$$

To prove this, we use a similar construction to that in Step 1. Let us first define attention weights by

$$
W_P = \begin{pmatrix} 0 & 0 & \mathbf{I}_d & 0 \\ 0 & 0 & 0 & 0 \\ 0 & 0 & 0 & 0 \\ 0 & 0 & 0 & 0 \end{pmatrix}, \quad W_Q = \begin{pmatrix} 0 & 0 & 0 & 0 \\ 0 & 0 & 0 & 0 \\ 0 & 0 & 0 & 0 \\ \mathbf{I}_d & 0 & 0 & 0 \end{pmatrix}.
$$

Then we have

$$
\begin{aligned}
W_P \cdot E_t E_t^T \cdot E_W e_t &= \begin{pmatrix} 0 & 0 & \mathbf{I}_d & 0 \\ 0 & 0 & 0 & 0 \\ 0 & 0 & 0 & 0 \\ 0 & 0 & 0 & 0 \end{pmatrix} \cdot \frac{1}{t} \sum_{i=1}^{t} \begin{pmatrix} a_i \\ a_i \\ x_i \\ x_{i-1} \end{pmatrix} \begin{pmatrix} a_i \\ a_i \\ x_i \\ x_{i-1} \end{pmatrix}^T \cdot \begin{pmatrix} 0 & 0 & 0 & 0 \\ 0 & 0 & 0 & 0 \\ 0 & 0 & 0 & 0 \\ \mathbf{I}_d & 0 & 0 & 0 \end{pmatrix} \begin{pmatrix} a_t \\ a_t \\ x_t \\ x_{t-1} \end{pmatrix} \\
&= \frac{1}{t} \sum_{i=1}^{t} \begin{pmatrix} x_i \\ 0 \\ 0 \\ 0 \end{pmatrix} \begin{pmatrix} x_{i-1}^T & 0^T & 0^T & 0^T \end{pmatrix} \begin{pmatrix} a_t \\ a_t \\ x_t \\ x_{t-1} \end{pmatrix} \\
&= \begin{pmatrix} \left( \frac{1}{t} \sum_{i=0}^{t-1} x_{i+1} x_i^T \right) a_t \\ 0 \\ 0 \\ 0 \end{pmatrix}.
\end{aligned}
$$

It follows that

$$
e_t + W_P \cdot E_t E_t^T \cdot E_W e_t = \begin{pmatrix} a_t + \left( \frac{1}{t} \sum_{i=0}^{t-1} x_{i+1} x_i^T \right) a_t \\ a_t \\ x_t \, x_{t-1} \end{pmatrix}.
$$

We then define $W_{MLP}$ to be any matrix such that

$$
\left( W_{MLP} \begin{pmatrix} a_t + \left( \frac{1}{t} \sum_{i=0}^{t-1} x_{i+1} x_i^T \right) a_t \\ a_t \\ x_t \, x_{t-1} \end{pmatrix} \right)_{1:d} = \left( \frac{1}{t} \sum_{i=0}^{t-1} x_{i+1} x_i^T \right) a_t.
$$

It then follows that, for $\widehat{e}_t = W_{MLP} \left( e_t + W_P \cdot E_t E_t^T \cdot W_Q e_t \right)$, we have

$$
(\widehat{e}_t)_{1:d} = \left( \frac{1}{t} \sum_{i=0}^{t-1} x_{i+1} x_i^T \right) a_t.
$$

**Step 4:** To conclude the proof, we compose the $L$-layer transformer constructed in step 2 with the single-layer transformer constructed in step 3 to construct an $(L+1)$-layer transformer which realizes the mapping

$$
e_t = \begin{pmatrix} 0 \\ 0 \\ x_t \\ x_{t-1} \end{pmatrix} \overset{\text{$L$ layers from Step 2}}{\longmapsto} \begin{pmatrix} z_{t,L} \\ z_{t,L} \\ x_t \\ x_{t-1} \end{pmatrix} \overset{\text{Single layer from Step 3}}{\longmapsto} \begin{pmatrix} \left( \frac{1}{t} \sum_{i=0}^{t-1} x_{i+1} x_i^T \right) z_{t,L} \\ * \\ * \\ * \end{pmatrix},
$$

where $*$ denotes entries which are not used in prediction. Since $\widehat{x}_{t+1,L} = \left( \frac{1}{t} \sum_{i=0}^{t-1} x_{i+1} x_i^T \right) z_{t,L}$, this proves the desired claim. $\qquad\square$

Before proving Theorem 1, we need to state some technical results about the dynamical system in Equation (2) and the least-squares estimator for its state matrix. First, we need bounds on the smallest and largest eigenvalues of the covariance matrix $X_T$. Such bounds are necessary for our proof, because the least squares estimator is not even well-defined unless the covariance is invertible, and in this case the condition number of $X_T$ governs the convergence rate of the associated Richardson iteration. To this end, we quote the following result from Foster et al. [2020]. Note that we have adapted the original result to our setting.

**Lemma 3.** *[Foster et al. [2020], Theorem 1] Let $(x_1, \ldots, x_t)$ be generated according to Equation* (2). *Then there is a numerical constant $c > 0$ such that, for any $\delta > 0$, as long as $T \geq cd\frac{\sigma^4}{(1-w_{\max}^2)^2} \log\left(\frac{d}{\delta(1+w_{\max}^2)} + 1\right)$, we have*

$$\frac{\sigma^2}{4} \cdot \mathbf{I}_d \prec X_t \prec \frac{4\sigma^2 d}{(1-w_{\max}^2)} \cdot \mathbf{I}_d \tag{9}$$

*with probability at least $1 - \delta$. In particular, as long as $T$ is sufficiently large, the bound in Equation (9) holds with probability $\frac{1}{2T^4}$.*

The second technical result, due to Matni and Tu [2019], provides a high-probability bound for the error of the least-squares estimator. This is crucial in bounding the test error achieved by the transformer constructed in Lemma 1.

**Lemma 4** (Matni and Tu [2019], Theorem 4.2)**.** *Let $(x_0, x_1, \ldots, x_t)$ denote the dynamical system described by Equation (2) and let $\widehat{W}_t$ denote the least-squares estimator (7) for the state matrix $W$. Then there exists a universal constant $c > 0$ such that whenever $T \geq \frac{cd\log(d/\delta)}{1-w_{\max}} \log\left(\frac{2\sigma^2}{1-w_{\max}^2}\right)$, we have*

$$\mathbb{E}\left\|W - \widehat{W}_T\right\|_{op}^2 \lesssim \frac{\sigma^2 d\log(d/\delta)}{T(1-w_{\max}^2)}.$$

*with probability at least $1 - \delta$. In particular, as long as $T$ is sufficiently large, we have*

$$\mathbb{E}\left\|W - \widehat{W}_T\right\|_{op}^2 \lesssim \frac{\sigma^2 (\log(T) + \log(4d))}{T(1-w_{\max}^2)} \tag{10}$$

*with probability at least $1 - \frac{1}{2T^4}$.*

We now give the proof of Theorem 1.

*Proof of Theorem 1.* We let $\widehat{y}_T$ denote the prediction of the $(L+1)$-layer transformer constructed in Lemma 1, where the step size $\alpha \in (0, 1)$ is independent of $T$ and $L$ and will be specified to satisfy some constraints later in the proof. Let $\mathcal{A}_T$ denote the event on which the bounds in Equations (9) and (10) hold, and assume that $T$ is sufficiently large that

$$\mathbb{P}\left(\mathcal{A}\right) \geq 1 - \frac{1}{T^4},$$

We can now write

$$\mathbb{E}\left\|\widehat{y}_T - Wx_T\right\|^2 = \mathbb{E}\left\|(\widehat{y}_T - Wx_T)\left(1_{\mathcal{A}_T} + 1_{\mathcal{A}_T^c}\right)\right\|^2$$
$$\leq 2\mathbb{E}\left\|(\widehat{y}_T - Wx_T)1_{\mathcal{A}_T}\right\|^2 + 2\mathbb{E}\left\|(\widehat{y}_T - Wx_T)1_{\mathcal{A}_T^c}\right\|^2$$

On the event $\mathcal{A}_T$, the least-squares estimator $\widehat{W}_T$ is well-defined, so we can bound the first expectation above by

$$\mathbb{E}\left\|(\widehat{y}_T - Wx_T)1_{\mathcal{A}_T}\right\| \lesssim 2\mathbb{E}\left\|\left(\widehat{y}_T - \widehat{W}_T x_T\right)1_{\mathcal{A}_T}\right\|^2 + 2\mathbb{E}\left\|\left(\widehat{W}_T - W\right)x_T \cdot 1_{\mathcal{A}_T}\right\|^2.$$

For the first term, we recall that the transformer prediction $\widehat{y}_T$ can be written as

$$\widehat{y}_T = \left(\frac{1}{T}\sum_{i=0}^{T-1} x_{i+1}x_i^T\right)z_{T,L},$$

where $z_{T,L}$ is the $L^{\text{th}}$ iterate of the Richardson method defined in Equation (8). In particular, Lemma 2 implies that

$$\|z_{T,L} - X_T^{-1}x_T\| \leq \|\mathbf{I}_d - \alpha X_T\|_{op}^L \|X_T^{-1}x_T\|.$$

Now, notice that $\|\mathbf{I}_d - \alpha X_T\|_{op} = \max\left(|1 - \alpha\lambda_{\max}(X_T)|, |1 - \lambda_{\min}(X_T)|\right)$. Since $\lambda_{\max}(X_T)$ and $\lambda_{\min}(X_T)$ are bounded on $\mathcal{A}_T$ according to Equation (9), we can choose $\alpha$ depending only on $d$, $\sigma^2$, and $w_{\max}$ to ensure that $\|\mathbf{I}_d - \alpha X_T\|_{op} \leq c_\alpha$ for some constant $c_\alpha \in (0, 1)$ whenever $\mathcal{A}_T$ holds. In particular, if we set

$$\alpha = \frac{8(1-w_{\max}^2)}{\sigma^2\left(16d + (1-w_{\max}^2)\right)},$$

then Equation (9) implies that

$$c_\alpha = \left(1 - \frac{4(1 - w_{\max}^2)}{16d + (1 - w_{\max}^2)}\right),$$

and $\|\mathbf{I}_d - \alpha X_T\| \le c_\alpha$ on $\mathcal{A}_T$. In turn, this allows us to bound

$$\mathbb{E}\left\|\left(\widehat{y}_T - \widehat{W}_T x_T\right) 1_{\mathcal{A}_T}\right\|^2 = \mathbb{E}\left\|\left(\frac{1}{T}\sum_{i=0}^{T-1} x_{i+1} x_i^T\right)\left(z_{T,L} - X_T^{-1} x_T\right) 1_{\mathcal{A}_T}\right\|^2$$

$$\le c_\alpha^{2L} \mathbb{E}\left\|\left(\frac{1}{T}\sum_{i=0}^{T-1} x_{i+1} x_i^T\right)\|X_T^{-1}\|_{\mathrm{op}}\|x_T\|\right\|^2$$

$$\le \frac{16 c_\alpha^{2L}}{\sigma^4} \mathbb{E}\left\|\left(\frac{1}{T}\sum_{i=0}^{T-1} x_{i+1} x_i^T\right)\|x_T\| \cdot 1_{\mathcal{A}_T}\right\|^2,$$

where we used that $\|X_T^{-1}\|_{\mathrm{op}} \le \frac{16}{\sigma^4}$ on $\mathcal{A}_T$. This gives the final bound

$$\mathbb{E}\left\|\left(\widehat{y}_T - \widehat{W}_T x_T\right) 1_{\mathcal{A}_T}\right\|^2 \le \frac{16}{\sigma^4} \cdot M_{1,\sigma,w_{\max}} \cdot c_\alpha^{2L}, \tag{11}$$

where we have defined

$$M_{1,\sigma,w_{\max}} := \sup_t \mathbb{E}\left[\left\|\frac{1}{T}\sum_{i=0}^{T-1} x_{i+1} x_i^T\right\|_{\mathrm{op}}^2 \|x_T\|^2\right].$$

Since the dynamical system is geometrically ergodic, the constant $M_{1,\sigma,w_{\max}}$ is finite and depends only on $\sigma^2$, $w_{\max}$, and $d$ (the techniques of Lemma 10 in Appendix C can be used to bound $M_{1,\sigma,w_{\max}}$ by the sixth moment of an appropriate Gaussian). To bound the difference between $\widehat{W}_T$ and $W$, we use Lemma 4, which gives the bound

$$2\mathbb{E}\left\|\left(\widehat{W}_T - W\right) x_T \cdot 1_{\mathcal{A}_T}\right\|^2 \lesssim \frac{\sigma^2 (\log(T) + \log(4d))}{T(1 - w_{\max}^2)} \mathbb{E}[\|x_T\|^2] \tag{12}$$

$$\le \frac{d\sigma^4 (\log(T) + \log(4d))}{T(1 - w_{\max}^2)^2}, \tag{13}$$

where we used Lemma 8 to bound $\mathbb{E}[\|x_T\|^2]$. We bound the expectation on $\mathcal{A}_T^c$ by the Cauchy-Schwarz inequality:

$$\mathbb{E}\left\|(\widehat{y}_T - W x_T) 1_{\mathcal{A}_T^c}\right\|^2 \le \mathbb{E}\left[\|\widehat{y}_T - W x_T\|^4\right]^{1/2} \cdot \mathbb{P}(\mathcal{A}_T^c)$$

$$\le 2\left(\mathbb{E}\|\widehat{y}_T\|^4 + \mathbb{E}\|W x_T\|^4\right)^{1/2} \cdot \mathbb{P}(\mathcal{A}_T^c)$$

$$\le 2\left(\mathbb{E}\left\|\left(\frac{1}{T}\sum_{i=0}^{T-1} x_{i+1} x_i^T\right) z_{T,L}\right\|^4 + \frac{d(d+2)\sigma^4}{(1 - w_{\max}^2)^2}\right)^{1/2} \cdot \mathbb{P}(\mathcal{A}_T^c)$$

$$\le 2\left(M_{2,\sigma,w_{\max}} \mathbb{E}[\|z_{T,L}\|^8]^{1/4} + \frac{\sqrt{d(d+2)}\sigma^2}{1 - w_{\max}^2}\right) \cdot \mathbb{P}(\mathcal{A}_T^c),$$

where

$$M_{2,\sigma,w_{\max}} := \sup_t \mathbb{E}\left[\left\|\frac{1}{T}\sum_{i=0}^{T-1} x_{i+1} x_i^T\right\|_{\mathrm{op}}\right]^{1/4}$$

is a constant depending only on $\sigma$, $w_{\max}$, and $d$. By Lemma 11, as long as $\alpha$ is small enough that $\alpha < \frac{2(1 - w_{\max}^2)}{\sigma^2}$, we have the following bound for $z_{T.L}$

$$\mathbb{E}[z_{T,L}\|^8]^{1/4} = O\left(1 + L^4 + T^{-2(L-1)} \cdot (16(L-1))!^{1/4}\right).$$

Note also that $\mathbb{P}(\mathcal{A}_T^c) = O\left(T^{-2}\right)$. This gives the bound

$$\mathbb{E}\left\|(\widehat{y}_T - Wx_T)\mathbb{1}_{\mathcal{A}_T^c}\right\|^2 \lesssim \left(M_{2,\sigma,w_{\max}}\left(1 + L^4 + T^{-2(L-1)} \cdot (16(L-1))!^{1/4}\right) + \frac{d\sigma^2}{1 - w_{\max}^2}\right)T^{-2}. \tag{14}$$

Combining Equations (11), (12), and (14), we have the final bound

$$\mathbb{E}\left\|\widehat{y}_T - Wx_T\right\|^2 \lesssim \frac{16}{\sigma^4} \cdot M_{1,\sigma,w_{\max}} \cdot c_\alpha^{2L} + \frac{d\sigma^4\left(\log(T) + \log(4d)\right)}{T(1 - w_{\max}^2)^2} \tag{15}$$

$$+ \left(M_{2,\sigma,w_{\max}}\left(1 + L^4 + T^{-2(L-1)} \cdot (16(L-1))!^{1/4}\right) + \frac{d\sigma^2}{1 - w_{\max}^2}\right)T^{-2}. \tag{16}$$

To balance the error between $L$ and $T$, we set

$$L = \frac{\log(T) + \log\left(\frac{16}{\sigma^4}M_{1,\sigma,w_{\max}}\right)}{2\log(1/c_\alpha)}.$$

Then it is clear that, when $T$ is sufficiently large, the bound

$$\left(M_{2,\sigma,w_{\max}}\left(1 + L^4 + T^{-2(L-1)} \cdot (16(L-1))!^{1/4}\right) + \frac{d\sigma^2}{1 - w_{\max}^2}\right)T^{-2} = O\left(\frac{\log(T)}{T}\right)$$

holds. This proves the final bound

$$\mathbb{E}\left\|\widehat{y}_T - Wx_T\right\|^2 \lesssim \frac{\log(T)}{T}, \tag{17}$$

for $T$ sufficiently large. The implicit constants depend on $\sigma$, $w_{\max}$, and $d$. Moreover, by tracking the implicit constants it can be seen that they depend only polynomially on $d$. Since the test loss is defined as

$$L_T(\theta) = \sup_{W \in \mathcal{W}} \mathbb{E}\left\|\widehat{y}_T - Wx_T\right\|^2,$$

the bound on $L_T$ as stated in Theorem 1 follows from taking the supremum over $\mathcal{W}$ on both sides of Equation (17). $\qquad\square$

## B  Proofs for Section 3.2

We break up the proof of Lemma 1 up into two separate lemmas. The first lemma, stated below, proves a formula for the pointwise limit of the individual loss function. The second lemma then proves the lower bound satisfied by the limiting loss. The proof of the first lemma involves several computations involving second, fourth, and sixth moments of the linear dynamical system. The proofs of these technical computations are deferred to Appendix C.

**Lemma 5.** *Consider the individual loss function*

$$\ell_T(\mathbf{p}, \mathbf{q}; w) = (\widehat{y}_T - wx_T)^2,$$

*where* $\mathbf{p}, \mathbf{q}$ *are the parameters of the linear attention block and* $w \in [w_{\min}, w_{\max}]$ *specifies the dynamical system defined by Equation* (5). *Then for any* $\mathbf{p}, \mathbf{q} \in \mathbb{R}^2$ *and* $w \in (0, 1)$, *we have*

$$\lim_{T \to \infty} \ell_T(\mathbf{p}, \mathbf{q}; w) = \frac{\sigma^2}{1 - w^2}\left(\frac{\sigma^2}{1 - w^2}(w\alpha_1 + \alpha_2) + w\right)^2 := \ell(\mathbf{p}, \mathbf{q}, w),$$

*where* $\alpha_1 = p_1q_2 + p_2q_1$ *and* $\alpha_2 = p_1q_1 + p_2q_2$.

*Proof.* We recall the formula for $\widehat{y}_t$:

$$\widehat{y}_T = \frac{1}{T}\sum_{i=1}^T \begin{pmatrix} p_1 & p_2 \end{pmatrix} \begin{pmatrix} \sum_{i=1}^T x_i^2 & \sum_{i=1}^T x_ix_{i-1} \\ \sum_{i=1}^T x_ix_{i-1} & \sum_{i=1}^T x_{i-1}^2 \end{pmatrix} \begin{pmatrix} q_1 \\ q_2 \end{pmatrix} x_T.$$

We aim to compute the limit of the expression

$$\mathbb{E}(\widehat{y}_T - wx_T)^2 = \mathbb{E}[\widehat{y}_T^2] + w^2\mathbb{E}[x_T^2] - 2w\mathbb{E}[\widehat{y}_Tx_T]. \tag{18}$$

**Step 1:** We first compute $\mathbb{E}[x_T^2]$. By Lemma 8, $x_T \sim N\left(0, \sigma^2 \frac{1-w^{2T}}{1-w^2}\right)$. We therefore have

$$\lim_{T\to\infty} \mathbb{E}[x_T^2] = \frac{\sigma^2}{1-w^2}. \tag{19}$$

**Step 2:** Next, we compute $\mathbb{E}[\widehat{y}_T x_T]$. From the formula $\widehat{y}_T$, we have

$$\widehat{y}_T x_T = p_1 q_1 \left(\frac{1}{T}\sum_{i=1}^T x_i^2 x_T^2\right) + p_2 q_2 \left(\frac{1}{T}\sum_{i=1}^T x_{i-1}^2 x_T^2\right) + (p_1 q_2 + p_2 q_1)\left(\frac{1}{T}\sum_{i=1}^T x_i x_{i-1} x_T^2\right).$$

By Lemma 9, we have

$$\lim_{T\to\infty} \widehat{y}_T x_T = \lim_{T\to\infty} p_1 q_1 \left(\frac{1}{T}\sum_{i=1}^T x_i^2 x_T^2\right) + p_2 q_2 \left(\frac{1}{T}\sum_{i=1}^T x_{i-1}^2 x_T^2\right) + (p_1 q_2 + p_2 q_1)\left(\frac{1}{T}\sum_{i=1}^T x_i x_{i-1} x_T^2\right)$$

$$= (p_1 q_1 + p_2 q_2)\frac{\sigma^4}{(1-w^2)^2} + (p_1 q_2 + p_2 q_1)\frac{\sigma^4 w}{(1-w^2)^2}$$

$$= \frac{\sigma^4}{(1-w^2)^2}\begin{pmatrix} p_1 & p_2 \end{pmatrix}\begin{pmatrix} 1 & w \\ w & 1 \end{pmatrix}\begin{pmatrix} q_1 \\ q_2 \end{pmatrix}.$$

**Step 3:** Next, we compute the limit of

$$\mathbb{E}[\widehat{y}_T^2] = \frac{1}{T^2}\mathbb{E}\left[\left(\begin{pmatrix} p_1 & p_2 \end{pmatrix}\begin{pmatrix} \sum_{i=1}^T x_i^2 & \sum_{i=1}^T x_i x_{i-1} \\ \sum_{i=1}^T x_i x_{i-1} & \sum_{i=1}^T x_{i-1}^2 \end{pmatrix}\begin{pmatrix} q_1 \\ q_2 \end{pmatrix}\right)^2 x_t^2\right]$$

$$= \frac{1}{T^2}\mathbb{E}\left[\left(p_1 q_1 \sum_{i=1}^T x_i^2 + (p_1 q_2 + q_1 p_2)\sum_{i=1}^T x_i x_{i-1} + p_2 q_2 \sum_{i=1}^T x_{i-1}^2\right)^2 x_T^2\right]$$

$$= \frac{1}{T^2}\mathbb{E}\left[\left(p_1 q_1 x_T^2 + (p_1 q_2 + q_1 p_2)\sum_{i=1}^T x_i x_{i-1} + (p_1 q_1 + p_2 q_2)\sum_{i=1}^T x_{i-1}^2\right)^2 x_T^2\right].$$

There are six terms to be computed. However, as we are only interested in when $T \to \infty$, then only terms with double sum survive. That is, we only need to compute $(\sum_{i=1}^T x_i x_{i-1})^2 x_T^2$, $(\sum_{i=1}^T x_{i-1}^2)^2 x_T^2$ and $(\sum_{i=1}^T x_i x_{i-1})(\sum_{i=1}^T x_{i-1}^2)x_T^2$. By Lemma 10, we have the computations

$$\lim_{t\to\infty} \frac{1}{T^2}\mathbb{E}\left[\left(\sum_{i=1}^T x_i x_{i-1}\right)^2 x_T^2\right] = \frac{\sigma^6 w^2}{(1-w^2)^3}$$

$$\lim_{T\to\infty} \frac{1}{T^2}\mathbb{E}\left[\left(\sum_{i=1}^T x_{i-1}^2\right)^2 x_T^2\right] = \frac{\sigma^6}{(1-w^2)^3}$$

$$\lim_{T\to\infty} \frac{1}{T^2}\mathbb{E}\left[\left(\sum_{i=1}^T x_i x_{i-1}\right)\left(\sum_{i=1}^T x_{i-1}^2\right)x_T^2\right] = \frac{\sigma^6 w}{(1-w^2)^3}.$$

Therefore,

$$\lim_{T\to\infty} \mathbb{E}[\widehat{y}_t^2] = \lim_{T\to\infty} \frac{1}{T^2}\mathbb{E}\left[\left(p_1 q_1 x_t^2 + (p_1 q_2 + q_1 p_2)\sum_{i=1}^T x_i x_{i-1} + (p_1 q_1 + p_2 q_2)\sum_{i=1}^T x_{i-1}^2\right)^2 x_T^2\right]$$

$$= \lim_{t\to\infty} \frac{1}{T^2}\mathbb{E}\left[\left((p_1 q_2 + p_2 q_1)^2\left(\sum_{i=1}^T x_i x_{i-1}\right)^2 + (p_1 q_1 + p_2 q_2)^2\left(\sum_{i=1}^T x_i x_{i-1}\right)^2\right.\right.$$

$$\left.\left. + 2(p_1 q_2 + p_2 q_1)(p_1 q_1 + p_2 q_2)\left(\sum_{i=1}^T x_i x_{i-1}\right)\left(\sum_{i=1}^T x_i x_{i-1}\right)\right)x_T^2\right]$$

$$= \frac{\sigma^6}{(1-w^2)^3}\left(\begin{pmatrix} p_1 & p_2 \end{pmatrix}\begin{pmatrix} 1 & w \\ w & 1 \end{pmatrix}\begin{pmatrix} q_1 \\ q_2 \end{pmatrix}\right)^2.$$

**Step 4:** Combining steps 1-3, we find that

$$\lim_{T\to\infty} \mathbb{E}[(\widehat{y}_T - wx_T)^2] = \frac{\sigma^6}{(1-w^2)^3}\left((p_1 \quad p_2)\begin{pmatrix} 1 & w \\ w & 1 \end{pmatrix}\begin{pmatrix} q_1 \\ q_2 \end{pmatrix}\right)^2$$

$$- 2w\frac{\sigma^4}{(1-w^2)^2}(p_1 \quad p_2)\begin{pmatrix} 1 & w \\ w & 1 \end{pmatrix}\begin{pmatrix} q_1 \\ q_2 \end{pmatrix} + \frac{\sigma^2}{1-w^2}$$

$$= \frac{\sigma^2}{1-w^2}\left(\frac{\sigma^2}{1-w^2}(w\alpha_1 + \alpha_2) - w\right)^2,$$

where we have defined $\alpha_1 = p_1q_2 + p_2q_1$ and $\alpha_2 = p_1q_1 + p_2q_2$. This concludes the proof. □

We now present the second lemma required to prove Lemma 1.

**Lemma 6.** *Let $\ell(\mathbf{p}, \mathbf{q}, w)$ denote the pointwise limit of the individual loss function, defined in Lemma 5. Then*

$$\inf_{\mathbf{p},\mathbf{q}} \sup_{w\in[w_{\min},w_{\max}]} \ell(\mathbf{p},\mathbf{q},w) \geq C(\sigma^2, w_{\min}, w_{\max}),$$

*where $C(\sigma^2, w_{\min}, w_{\max}) > 0$ is a strictly positive constant which depends only on $\sigma^2$, $w_{\min}$, and $w_{\max}$.*

*Proof.* We recall that $\ell(\mathbf{p}, \mathbf{q}, w) = \frac{\sigma^2}{1-w^2}\left(\frac{\sigma^2}{1-w^2}(w\alpha_1 + \alpha_2) - w\right)^2$, where $\alpha_1$ and $\alpha_2$ are related to $\mathbf{p}$ and $\mathbf{q}$ by $\alpha_1 = p_1q_2 + p_2q_1$ and $\alpha_2 = p_1q_1 + p_2q_2$. We therefore aim to show that

$$\inf_{\mathbf{p},\mathbf{q}} \sup_{w\in[w_{\min},w_{\max}]} \frac{\sigma^2}{1-w^2}\left(\frac{\sigma^2}{1-w^2}(w\alpha_1 + \alpha_2) + w\right)^2 \geq C(\sigma^2, w_{\min}, w_{\max}),$$

where $C(\sigma^2, w_{\min}, w_{\max}) > 0$ is a strictly positive constant. Clearly, we have

$$\inf_{\mathbf{p},\mathbf{q}} \sup_{w\in[w_{\min},w_{\max}]} \frac{\sigma^2}{1-w^2}\left(\frac{\sigma^2}{1-w^2}(w\alpha_1 + \alpha_2) - w\right)^2$$

$$\geq \inf_{\alpha_1,\alpha_2} \sup_{w\in[w_{\min},w_{\max}]} \frac{\sigma^2}{1-w^2}\left(\frac{\sigma^2}{1-w^2}(w\alpha_1 + \alpha_2) - w\right)^2,$$

where the infimum on the right is taken over all $(\alpha_1, \alpha_2) \in \mathbb{R}^2$. We abuse notation and denote the above function by $\ell(\alpha_1, \alpha_2; w)$. Note that we also have the lower bound

$$\inf_{\alpha_1,\alpha_2} \sup_{w\in[w_{\min},w_{\max}]} \ell(\alpha_1, \alpha_2; w) \tag{20}$$

$$\geq \inf_{\alpha_1,\alpha_2} \max\left(\ell(\alpha_1, \alpha_2; w_{\min}), \ell(\alpha_1, \alpha_2; w_{\max}), \ell\left(\alpha_1, \alpha_2, \frac{w_{\min} + w_{\max}}{2}\right)\right). \tag{21}$$

The infimum on the right is attained at a point $(\alpha_1^*, \alpha_2^*)$ on the intersection of the curves $\ell(\cdot, \cdot; w_{\min})$, $\ell(\cdot, \cdot; w_{\max})$ and $\ell\left(\cdot, \cdot, \frac{w_{\min}+w_{\max}}{2}\right)$, and it is equal to zero if and only if there exist parameters $\alpha_1^*, \alpha_2^*$ such that $\ell(\alpha_1^*, \alpha_2^*; w) = 0$ for all $w \in \left\{w_{\min}, w_{\max}, \frac{w_{\min}+w_{\max}}{2}\right\}$. For a fixed $w$, a point $(\alpha_1, \alpha_2)$ satisfies $\ell(\alpha_1, \alpha_2; w) = 0$ if and only if it lies on the line

$$\frac{\sigma^2 w}{1-w^2}x + \frac{\sigma^2}{1-w^2}y = w.$$

Since $w_{\min} \neq w_{\max}$, it is easy to see, for the system of equations

$$\begin{cases} \frac{\sigma^2 w_{\min}}{1-w_{\min}^2}x + \frac{\sigma^2}{1-w_{\min}^2}y = w_{\min} \\ \frac{\sigma^2 w_{\max}}{1-w_{\max}^2}x + \frac{\sigma^2}{1-w_{\max}^2}y = w_{\max} \\ \frac{\sigma^2 \tilde{w}}{1-\tilde{w}^2}x + \frac{\sigma^2}{1-\tilde{w}^2}y = \tilde{w}, \quad \tilde{w} = \frac{1}{2}(w_{\min} + w_{\max}), \end{cases}$$

that the rank of the augmented matrix is strictly greater than the rank of the coefficient matrix. By the Rouche-Capelli Theorem Shafarevich and Remizov [2012], the above system has no solution. This proves that the infimum

$$\inf_{\alpha_1,\alpha_2} \max\left(\ell(\alpha_1, \alpha_2; w_{\min}), \ell(\alpha_1, \alpha_2; w_{\max}), \ell\left(\alpha_1, \alpha_2, \frac{w_{\min} + w_{\max}}{2}\right)\right) = C(\sigma^2, w_{\min}, w_{\max})$$

is nonzero, and hence the desired claim is proven. □

We now leverage Lemmas 5 and 6 to prove that the limit of the infimum $\inf_{\mathbf{p},\mathbf{q}} L_T(\mathbf{p},\mathbf{q})$ is lower bounded away from zero. We first need another lemma. Recall that a family of functions is *equicontinuous* if it has a uniform modulus of continuity (see Appendix C for a precise definition). The following lemma establishes some continuity properties of the sequence of individual losses $\{\ell_T\}$.

**Lemma 7.**      *1. For any $\mathbf{p},\mathbf{q} \in \mathbb{R}^2$, the sequence of functions on $[w_{\min}, w_{\max}]$ given by $\{w \mapsto \ell_T(\mathbf{p},\mathbf{q},w)\}_{T=2}^{\infty}$ is equicontinuous.*

     *2. For any $R > 0$, the family of functions on $\{\|\mathbf{p}\|, \|\mathbf{q}\| \leq R\}$ given by*

$$\{(\mathbf{p},\mathbf{q}) \mapsto \ell_T(\mathbf{p},\mathbf{q},w)\}_{T \geq 2, \, w \in [w_{\min}, w_{\max}]}$$

     *is equicontinuous.*

*Proof.* We first recall that $\ell_T(\mathbf{p},\mathbf{q},w) = \mathbb{E}[(\widehat{y}_T - wx_T)^2]$.

For 1), the expectations defining $\ell_T$ are computed in Lemmas 9 and 10, and it is easily seen that, for fixed $\mathbf{p},\mathbf{q}$, the derivative in $w$ of each term of $\ell_T(\mathbf{p},\mathbf{q},w)$ is uniformly bounded for $w \in [w_{\min}, w_{\max}]$.

For 2), we note that for each $T$ and $w \in [w_{\min}, w_{\max}]$, $\ell_T(\mathbf{p},\mathbf{q},w)$ is a polynomial in the entries of $\mathbf{p}$ and $\mathbf{q}$, and the computations of Lemmas 9 and 10 show that the coefficients of these polynomials are uniformly bounded in $T$ and $w$. This proves that the derivatives of the functions $\{(\mathbf{p},\mathbf{q}) \mapsto \ell_T(\mathbf{p},\mathbf{q},w)\}_{T \geq 2, \, w \in [w_{\min}, w_{\max}]}$ are uniformly bounded with respect to $T$ and $w$ on $\{\|\mathbf{p}\|, \|\mathbf{q}\| \leq R\}$. $\qquad\square$

We are now ready to prove Theorem 2.

*Proof of Theorem 2.* By definition, we have

$$L_T(\mathbf{p},\mathbf{q}) = \sup_{w \in [w_{\min}, w_{\max}]} \ell_T(\mathbf{p},\mathbf{q},w).$$

The result will be established if we can prove that

$$\lim_{T \to \infty} \inf_{\|\mathbf{p}\|, \|\mathbf{q}\| \leq R} \sup_{w \in [w_{\min}, w_{\max}]} \ell_T(\mathbf{p},\mathbf{q},w) = \inf_{\|\mathbf{p}\|, \|\mathbf{q}\| \leq R} \sup_{w \in [w_{\min}, w_{\max}]} \ell(\mathbf{p},\mathbf{q},w), \qquad (22)$$

where $\ell$ is the pointwise limit of $\ell_T$, defined in Lemma 5. Once Equation (22) has been established, the result follows from Theorem 2, which establishes that

$$\inf_{\|\mathbf{p}\|, \|\mathbf{q}\| \leq R} \sup_{w \in [w_{\min}, w_{\max}]} \ell(\mathbf{p},\mathbf{q},w) \geq C(\sigma^2, w_{\min}, w_{\max}).$$

To prove Equation (22), we first note that for any $\mathbf{p},\mathbf{q} \in \mathbb{R}^2$ the sequence of functions on $[w_{\min}, w_{\max}]$ given by $\{w \mapsto \ell_T(\mathbf{p},\mathbf{q},w)\}_T$ is equicontinuous by Lemma 7 and converges pointwise to the function $w \mapsto \ell(\mathbf{p},\mathbf{q},w)$ by Lemma 5. Therefore, by Lemma 13, the sequence of functions $\{w \mapsto \ell_T(\mathbf{p},\mathbf{q},w)\}_T$ converges uniformly on $[w_{\min}, w_{\max}]$ to the function $w \mapsto \ell(\mathbf{p},\mathbf{q},w)$. In particular, we have

$$\lim_{T \to \infty} \sup_{w \in [w_{\min}, w_{\max}]} \ell_T(\mathbf{p},\mathbf{q},w) = \sup_{w \in [w_{\min}, w_{\max}]} \ell(\mathbf{p},\mathbf{q},w), \quad \text{for all } \mathbf{p},\mathbf{q} \in \mathbb{R}^2. \qquad (23)$$

Next, Lemma 7 guarantees that the sequence of functions on $\{\|\mathbf{p}\|, \mathbf{q}\| \leq R\}$ defined by $\{(\mathbf{p},\mathbf{q}) \mapsto \ell_T(\mathbf{p},\mathbf{q},w)\}_{T,w}$ is equicontinuous. Since the modulus of continuity of a family of functions is preserved under the supremum, the family of functions $\{(\mathbf{p},\mathbf{q}) \mapsto \sup_{w \in [w_{\min}, w_{\max}]} \ell_T(\mathbf{p},\mathbf{q},w)\}$ is also equicontinuous $\{\|\mathbf{p}\|, \mathbf{q}\| \leq R\}$. Therefore, Equation 23 and another application of Lemma 13 together imply that

$$\sup_{w \in [w_{\min}, w_{\max}]} \ell_T(\mathbf{p},\mathbf{q},w) \to \sup_{w \in [w_{\min}, w_{\max}]} \ell(\mathbf{p},\mathbf{q},w), \quad \text{uniformly on } \{\|\mathbf{p}\|, \mathbf{q}\| \leq R\}.$$

This proves Equation (22) and concludes the proof. $\qquad\square$

## C  Auxiliary lemmas

In this section, we state and prove various helpful lemmas used in the proofs of Theorems 1 and 2. To begin, make frequent use of the following characterization of the distribution of the dynamical system iterates defined by Equation (2).

**Lemma 8.** *Let $\mathbf{x} = (x_0, x_1, \ldots, x_T)$ be as defined in Equation (2).*

1. *For any $t \in \{1, \ldots, T\}$, $x_t \sim N\left(0, \sigma^2 (\mathbf{I}_d - W^{2t})(\mathbf{I}_d - W^2)^{-1}\right)$.*

2. *For any $1 \le i < j \le T$, the random variable $z_{i,j} = \sum_{k=i+1}^{j} W^{j-k} \xi_k$ satisfies $z_{i,j} \perp x_i$, $\mathbb{E}[z_{i,j}] = 0$, and $x_j = W^{j-i} x_i + z_{i,j}$. In addition, $z_{i,j}$ is a normal random variable with covariance $\sigma^2 \sum_{k=i+1}^{j} W^{2(j-k)}$*

*Proof.* 1) For any $t$, $x_t = \sum_{i=1}^{t} W^{t-i} \xi_i$. It follows that

$$x_t \sim N\left(0, \sigma^2 \sum_{i=1}^{t} W^{2(t-i)}\right) = N\left(0, \sigma^2 (\mathbf{I}_d - W^{2t})(\mathbf{I}_d - W^2)^{-1}\right).$$

2) From the above calculation, it is clear that $x_j - W^{j-i} x_i = z_{i,j}$. The independence of $z_{i,j}$ and $x_i$ follows from the independence of $\xi_1, \ldots, \xi_T$, and the fact that $\mathbb{E}[z_{i,j}] = 0$ follows from the fact that $\mathbb{E}[\xi_k] = 0$ for each $k \in \{1, \ldots, T\}$. $\qquad\square$

The following computations, involving fourth and sixth moments of the linear dynamical system, are heavily used in the proof of Lemma 5.

**Lemma 9.** *[Fourth moments along the dynamics] Let $(x_1, \ldots, x_t)$ be the iterates of the 1D dynamical system defined by Equation (5) with parameter $w \in (0, 1)$. Then*

1. $\lim_{t \to \infty} \frac{1}{t} \sum_{i=1}^{t} \mathbb{E}[x_i x_{i-1} x_t] = \frac{\sigma^4 w}{(1-w^2)^2}$.

2. $\lim_{t \to \infty} \frac{1}{t} \sum_{i=1}^{t} \mathbb{E}[x_i^2 x_t^2] = \frac{\sigma^4}{(1-w^2)^2} = \lim_{t \to \infty} \frac{1}{t} \sum_{i=1}^{t} \mathbb{E}[x_{i-1}^2 x_t^2]$.

*Proof.* For the first equation, we first compute $\mathbb{E}[x_i x_{i-1} x_t^2]$ for each $t > 0$ and $i \in \{1, \ldots, t\}$. For indices $i < j$, we use the decomposition $x_j = w^{j-1} x_i + z_{i,j}$ as defined in Lemma 8. We have

$$
\begin{aligned}
\mathbb{E}[x_i x_{i-1} x_t^2] &= \mathbb{E}[x_i x_{i-1} (w^{t-i} x_i + z_{i,t})^2] \\
&= \mathbb{E}[w^{2(t-i)} x_i^3 x_{i-1} + x_i x_{i-1} z_{i,t}^2] \\
&= \mathbb{E}[w^{2(t-i)} (w x_{i-1} + \xi_i)^3 x_{i-1} + x_i x_{i-1} z_{i,t}^2] \\
&= \mathbb{E}[w^{2(t-i)} (w^3 x_{i-1}^4 + 3 w x_{i-1}^2 \xi_i^2) + x_i x_{i-1} z_{i,t}^2] \\
&= 3\sigma^4 w^{2(t-i)+3} \left(\frac{1 - w^{2(i-1)}}{1 - w^2}\right)^2 + 3\sigma^4 w^{2(t-i)+1} \frac{1 - w^{2(i-1)}}{1 - w^2} + \sigma^4 w \frac{1 - w^{2(i-1)}}{1 - w^2} \frac{1 - w^{2(t-i)}}{1 - w^2} \\
&= \sigma^4 \left(3 w^{2(t-i)+3} \frac{1 - 2w^{2(i-1)} + w^{4(i-1)}}{(1 - w^2)^2} + 3 w^{2(t-i)+1} \frac{1 - w^{2(i-1)}}{1 - w^2}\right. \\
&\quad \left. + w \frac{1 - w^{2(i-1)} - w^{2(t-i)} + w^{2(t-1)}}{(1 - w^2)^2}\right) \\
&= \sigma^4 \left(\frac{3 w^{2(t-i)+3} - 6 w^{2(t-1)+3} + 3 w^{2(t+i)-1}}{(1 - w^2)^2} + \frac{3 w^{2(t-i)+1} - 3 w^{2(t-1)+1}}{1 - w^2}\right) \\
&\quad + \sigma^4 \left(\frac{w - w^{2(i-1)+1} - w^{2(t-i)+1} + w^{2(t-1)+1}}{(1 - w^2)^2}\right)
\end{aligned}
$$

Notice that we are able to compute the above expectations easily because many of the terms vanish due to independence and the fact that all distinct random variables are centered. Summing over $t$, we get

$$\sum_{i=1}^{t} \mathbb{E}[x_i x_{i-1} x_t^2] = \sigma^4 \frac{3}{(1-w^2)^2} \left( w^3 \frac{1-w^{2t}}{1-w^2} - 2tw^{2(t-1)+3} + w^{2(t+1)-1} \frac{1-w^{2t}}{1-w^2} \right)$$

$$+ \sigma^4 \frac{3}{1-w^2} \left( w \frac{1-w^{2t}}{1-w^2} - tw^{2(t-1)+1} \right)$$

$$+ \sigma^4 \frac{w}{(1-w^2)^2} \left( t - \frac{1-w^{2t}}{1-w^2} - \frac{1-w^{2t}}{1-w^2} + tw^{2(t-1)} \right)$$

$$= \sigma^4 \frac{1-w^{2t}}{1-w^2} \left( \frac{3w^3}{(1-w^2)^2} + \frac{3w^{2(t+1)-1}}{(1-w^2)^2} + \frac{3w}{1-w^2} - 2\frac{w}{(1-w^2)^2} \right)$$

$$+ \sigma^4 t \left( -\frac{6w^{2(t-1)+3}}{(1-w^2)^2} - \frac{3w^{2(t-1)+1}}{1-w^2} + \frac{w}{(1-w^2)^2} + \frac{w^{2(t-1)+1}}{(1-w^2)^2} \right)$$

$$= \sigma^4 \frac{1-w^{2t}}{1-w^2} \frac{3w^3 + 3w^{2(t+1)-1} + 3w(1-w^2) - 2w}{(1-w^2)^2}$$

$$+ \sigma^4 t \frac{-6w^{2(t-1)+3} - 3w^{2(t-1)+1}(1-w^2) + w + w^{2(t-1)+1}}{(1-w^2)^2}$$

$$= \sigma^4 \frac{1-w^{2t}}{1-w^2} \frac{3w^{2(t+1)-1} + w}{(1-w^2)^2} + \sigma^4 t \frac{-3w^{2(t-1)+3} - 2w^{2(t-1)+1} + w}{(1-w^2)^2}.$$

We then divide by $t$ to get

$$\frac{1}{t} \sum_{i=1}^{t} \mathbb{E}[x_i x_{i-1} x_t^2] = \sigma^4 \frac{-3w^{2(t-1)+3} - 2w^{2(t-1)+1} + w}{(1-w^2)^2} + O\left( \frac{1}{t} \right).$$

Since $w \in (0,1)$, all terms of the form $w^{at+b}$ vanish when we take the limit. This gives us the final result

$$\lim_{t \to \infty} \frac{1}{t} \sum_{i=1}^{t} \mathbb{E}[x_i x_{i-1} x_t^2] = \frac{\sigma^2 w}{(1-w^2)^2}. \tag{24}$$

For the second equation, we first prove that

$$\lim_{t \to \infty} \frac{1}{t} \sum_{i=1}^{t} \mathbb{E}[x_i^2 x_t^2] = \frac{\sigma^4}{(1-w^2)^2}.$$

To prove this, we first rewrite the expression at hand as

$$\frac{1}{t} \sum_{i=1}^{t} \mathbb{E}[x_i^2 x_t^2] = \frac{1}{t} \sum_{i=1}^{t} \mathbb{E}[x_i (w x_{i-1} + \xi_i) x_t^2]$$

$$= w \cdot \frac{1}{t} \sum_{i=1}^{t} \mathbb{E}[x_i x_{i-1} x_t^2] + \frac{1}{t} \sum_{i=1}^{t} \mathbb{E}[x_i \xi_i x_t^2]$$

To simplify the second sum on the right, we further compute

$$\mathbb{E}[x_i \xi_i x_t^2] = \mathbb{E}[x_i \xi_i (w^{t-i} x_i + z_{i,t})^2]$$

$$= \mathbb{E}[w^{2(t-i)} x_i^3 \xi_i + x_i \xi_i z_{i,t}^2]$$

$$= \mathbb{E}[w^{2(t-i)+3} \xi_i^4 + \xi_i^2 z_{i,t}^2]$$

$$= \sigma^4 \left( 3w^{2(t-i)+3} + \frac{1-w^{2(t-i)}}{1-w^2} \right).$$

This allows us to compute the limit

$$\lim_{t\to\infty} w \cdot \frac{1}{t}\sum_{i=1}^{t}\mathbb{E}[x_i x_{i-1} x_t^2] + \frac{1}{t}\sum_{i=1}^{t}\mathbb{E}[x_i \xi_i x_t^2] = \frac{\sigma^4 w^2}{(1-w^2)^2} + \frac{\sigma^4}{1-w^2}$$

$$= \frac{\sigma^4}{(1-w^2)^2}.$$

This proves that

$$\lim_{t\to\infty}\frac{1}{t}\sum_{i=1}^{t}\mathbb{E}[x_i^2 x_t^2] = \frac{\sigma^4}{(1-w^2)^2},$$

as desired. To prove that the sum $\frac{1}{t}\sum_{i=1}^{t}\mathbb{E}[x_{i-1}^2 x_t^2]$ has the same limit, we observe by a similar computation that

$$\frac{1}{t}\sum_{i=1}^{t}\mathbb{E}[x_i^2 x_t^2] = \frac{1}{t}\sum_{i=1}^{t}\mathbb{E}[(wx_{i-1}+\xi_i)x_t^2]$$

$$= w^2\frac{1}{t}\sum_{i=1}^{t}\mathbb{E}[x_{i-1}^2 x_t^2] + \frac{1}{t}\sum_{i=1}^{t}\mathbb{E}[\xi_i^2 x_t^2]$$

$$= w^2\frac{1}{t}\sum_{i=1}^{t}\mathbb{E}[x_{i-1}^2 x_t^2] + \frac{1}{t}\sum_{i=1}^{t}\mathbb{E}[\xi_i^2 (w^{t-i}x_i + z_{i,t})^2]$$

$$= w^2\frac{1}{t}\sum_{i=1}^{t}\mathbb{E}[x_{i-1}^2 x_t^2] + \frac{1}{t}\sum_{i=1}^{t}\mathbb{E}[w^{2(t-i)}\xi_i^4 + \xi_i^2 z_{i,t}^2]$$

$$= w^2\frac{1}{t}\sum_{i=1}^{t}\mathbb{E}[x_{i-1}^2 x_t^2] + \frac{1}{t}\sum_{i=1}^{t}\sigma^4\left(3w^{2(t-i)} + \frac{1-w^{2(t-i)}}{1-w^2}\right).$$

Rearranging terms and taking limits $t\to\infty$ on both sides, we get that

$$\lim_{t\to\infty}\frac{1}{t}\sum_{i=1}^{t}\mathbb{E}[x_{i-1}^2 x_i^2] = w^{-2}\left(\frac{\sigma^4}{(1-w^2)^2} - \frac{1}{1-w^2}\right)$$

$$= \frac{\sigma^4}{(1-w^2)^2}.$$

This completes the proof. $\qquad\square$

**Lemma 10.** *[Sixth moments along the dynamics] Let* $(x_1,\ldots,x_t)$ *be the iterates of the 1D dynamical system defined by Equation* (5) *with parameter* $w\in(0,1)$. *Then*

1. $\lim_{t\to\infty}\frac{1}{t^2}\mathbb{E}\left[\left(\sum_{i=1}^{t}x_i x_{i-1}\right)^2 x_t^2\right] = \frac{\sigma^6 w^2}{(1-w^2)^3}.$

2. $\lim_{t\to\infty}\frac{1}{t^2}\mathbb{E}\left[\left(\sum_{i=1}^{t}x_{i-1}^2\right)^2 x_t^2\right] = \frac{\sigma^6}{(1-w^2)^3}.$

3. $\lim_{t\to\infty}\frac{1}{t^2}\mathbb{E}\left[\left(\sum_{i=1}^{t}x_i x_{i-1}\right)\left(\sum_{i=1}^{t}x_{i-1}^2\right)x_t^2\right] = \frac{\sigma^6 w}{(1-w^2)^3}.$

*Proof.* 1) For the first equation, write

$$\frac{1}{t^2}\mathbb{E}\left[\left(\sum_{i=1}^{t}x_i x_{i-1}\right)^2 x_t^2\right] = \frac{1}{t^2}\mathbb{E}\left[\sum_{i=1}^{t}x_i^2 x_{i-1}^2 x_t^2 + 2\sum_{j=2}^{t}\sum_{i=1}^{j-1}x_i x_{i-1}x_j x_{j-1}x_t^2\right].$$

Notice, however, that $\frac{1}{t^2}\sum_{i=1}^{t}\mathbb{E}[x_i^2 x_{i-1}^2 x_t^2] = O(t^{-1})$, and thus it suffices to only consider the second sum to compute the limit. For each pair $(i,j)$, the expectation of $x_i x_{i-1} x_j x_{j-1} x_t^2$ decomposes into a sum of many different terms, so computing the precise value of the limit of the sum

$\sum_{j=2}^{t}\sum_{i=1}^{j-1} x_i x_{i-1} x_j x_{j-1} x_t^2$ is ostensibly challenging. However, notice that any term $a_{i,j}$ such that $\sum_{j=2}^{t}\sum_{i=1}^{j-1} a_{i,j} = o\left(t^2\right)$ will vanish when we take the limit. In particular, since $\sum_k w^k < \infty$, any term of the form $a_{i,j} = c \cdot w^{a(t-j)+b(j-i)+ci+d}$ where $c$ is a constant, *will not contribute to the limit*. This allows us to ignore most terms arising from the subsequent expectations

To proceed with the proof, we compute for $i < j$ that

$$\mathbb{E}[x_i x_{i-1} x_j x_{j-1} x_t^2] = \mathbb{E}\left[(w^{t-j} x_j + z_{j,t})^2 x_i x_{i-1} x_j x_{j-1}\right]$$

$$= \mathbb{E}\left[w^{2(t-j)} x_j^2 x_i x_{i-1} x_j x_{j-1}\right] + \mathbb{E}\left[z_{j,t}^2 x_i x_{i-1} x_j x_{j-1}\right] = I + II,$$

where we used the independence and mean-zero property of $z_{j,t}$. For the first term, we get that

$$\mathbb{E}\left[w^{2(t-j)} x_j^2 x_i x_{i-1} x_j x_{j-1}\right] = w^{2(t-j)} \mathbb{E}\left[x_j^3 x_{j-1} x_i x_{i-1}\right]$$

$$= w^{2(t-j)} \mathbb{E}\left[(w x_{j-1} + \xi_j)^3 x_{j-1} x_i x_{i-1}\right]$$

$$= w^{2(t-j)} \mathbb{E}\left[\left(w^3 x_{j-1}^3 + 3 w x_{j-1} \xi_j^2\right) x_{j-1} x_i x_{i-1}\right]$$

$$= w^{2(t-j)} \mathbb{E}\left[w^3 x_{j-1}^4 x_i x_{i-1} + 3\sigma^2 w x_{j-1}^2 x_i x_{i-1}\right]$$

$$= w^{2(t-j)} \mathbb{E}\left[w^3 (w^{j-i-1} x_i + z_{i,j-1})^4 x_i x_{i-1} + 3\sigma^2 (w^{j-i-1} x_i + z_{i,j-1})^2 x_i x_{i-1}\right]$$

$$= w^{2(t-j)} \mathbb{E}\left[w^3 \left(w^{4(j-i-1)} x_i^4 + 6 w^{2(j-i-1)} x_i^2 z_{i,j-1}^2 + z_{i,j-1}^4\right) x_i x_{i-1} + 3\sigma^2 w \left(w^{2(j-i-1)} x_i^2 + z_{i,j-1}^2\right) x_i x_{i-1}\right]$$

$$= w^{2(t-j)} \mathbb{E}\left[w^3 \left(w^{4(j-i-1)}\left(w x_{i-1} + \xi_i\right)^5 x_{i-1} + 6 w^{2(j-i-1)} z_{i,j-1}^2 \left(w x_{i-1} + \xi_i\right)^3 x_{i-1}\right.\right.$$

$$\left.+ z_{i,j-1}^4 \left(w x_{i-1} + \xi_i\right) x_{i-1}\right)\Bigg]$$

$$+ 3 w^{2(t-j)} \mathbb{E}\left[\sigma^2 w \left(w^{2(j-i-1)}\left(w x_{i-1} + \xi_i\right)^3 x_{i-1} + z_{i,j-1}^2 \left(w x_{i-1} + \xi_i\right) x_{i-1}\right)\right]$$

$$= w^{2(t-j)} w^3 w^{4(j-i-1)} \mathbb{E}\left[w^5 x_{i-1}^6 + 10 w^3 \xi_i^2 x_{i-1}^4 + 5 w \xi_i^4 x_{i-1}^2\right]$$

$$+ 6 w^{2(t-j)} w^3 w^{2(j-i-1)} \mathbb{E}[z_{i,j-1}^2] \mathbb{E}\left[w^3 x_{i-1}^4 + 3 \xi_i^2 w^2 x_{i-1}^2\right]$$

$$+ w^{2(t-j)} w^3 \mathbb{E}[z_{i,j-1}^4] \mathbb{E}\left[w x_{i-1}^2\right] + 3\sigma^2 w^{2(t-j)+1} \mathbb{E}\left[w^{2(j-i-1)}\left(w^3 x_{i-1}^4 + 3\xi_i^2 w^2 x_{i-1}^2\right)\right]$$

$$+ 3\sigma^2 w^{2(t-j)+1} \mathbb{E}[z_{i-j-1}^2] \mathbb{E}\left[w x_{i-1}^2\right]$$

$$= w^{2(t-j)} w^3 w^{4(j-i-1)} \sigma^6 \left(15 w^5 \left(\frac{1-w^{2i}}{1-w^2}\right)^3 + 30 w^3 \left(\frac{1-w^{2i}}{1-w^2}\right)^2 + 15 w \left(\frac{1-w^{2i}}{1-w^2}\right)\right)$$

$$+ 6 w^{2(t-j)} w^3 w^{2(j-i-1)} \sigma^6 \left(\frac{1-w^{2(j-i)}}{1-w^2}\right)\left(3 w^3 \left(\frac{1-w^{2i}}{1-w^2}\right)^2 + 3 w^2 \left(\frac{1-w^{2i}}{1-w^2}\right)\right)$$

$$+ 3 w^{2(t-j)+1} w^3 \sigma^6 \left(\frac{1-w^{2(j-i)}}{1-w^2}\right)^2 \left(\frac{1-w^{2i}}{1-w^2}\right)$$

$$+ 3 w^{2(t-j)+1} w^{2(j-i-1)} \sigma^6 \left(3 w^3 \left(\frac{1-w^{2i}}{1-w^2}\right)^2 + 3 w^2 \left(\frac{1-w^{2i}}{1-w^2}\right)\right) + 3 w^{2(t-j)+2} \sigma^6 \left(\frac{1-w^{2(j-i)}}{1-w^2}\right)\left(\frac{1-w^{2i}}{1-w^2}\right).$$

As discussed previously, since each of the above terms is $o(t^2)$, all of them vanish when we take the limit; that is,

$$\lim_{t\to\infty} \frac{1}{t^2} \sum_{j=2}^{t}\sum_{i=1}^{j-1} \mathbb{E}\left[w^{2(t-j)} x_j^3 x_i x_{i-1} x_{j-1}\right] = 0.$$

For the second term, we compute

$$\mathbb{E}\left[z_{j,t}^2 x_i x_{i-1} x_j x_{j-1}\right] = \mathbb{E}[z_{j,t^2}]\mathbb{E}[wx_{j-1}^2 x_i x_{i-1}]$$
$$= \mathbb{E}[z_{j,t}^2]\mathbb{E}[w(w^{j-i-1}x_i + z_{i,j-1})^2 x_i x_{i-1}]$$
$$= \mathbb{E}[z_{j,t}^2]\mathbb{E}[w^{j-i}x_i^3 x_{i-1} + wz_{i,j-i}^2 x_i x_{i-1}]$$
$$= \mathbb{E}[z_{j,t}^2]\mathbb{E}[w^{j-i+1}x_{i-1}^4 + w^2 z_{i,j-1}^2 x_{i-1}^2]$$
$$= w^{j-i+1}\mathbb{E}[z_{j,t}^2]\mathbb{E}[x_{i-1}^4] + w^2\mathbb{E}[z_{j,t}^2]\mathbb{E}[z_{i,j-1}^2]\mathbb{E}[x_{i-1}^2]$$
$$= \sigma^6 w^{j-i+1}\left(\frac{1-w^{2(t-j)}}{1-w^2}\right)\left(\frac{1-w^{2i}}{1-w^2}\right)^2 + \sigma^6 w^2\left(\frac{1-w^{2(t-j)}}{1-w^2}\right)\left(\frac{1-w^{2(j-i)}}{1-w^2}\right)\left(\frac{1-w^{2i}}{1-w^2}\right).$$

The first term $\sigma^6 w^{j-i+1}\left(\frac{1-w^{2(t-j)}}{1-w^2}\right)\left(\frac{1-w^{2i}}{1-w^2}\right)^2$ is $o(t^2)$ and hence vanishes in the limit. The second term satisfies

$$\lim_{t\to\infty}\frac{2}{t^2}\sum_{j=2}^{\infty}\sum_{i=1}^{t-1}\sigma^6 w^2\left(\frac{1-w^{2(t-j)}}{1-w^2}\right)\left(\frac{1-w^{2(j-i)}}{1-w^2}\right)\left(\frac{1-w^{2i}}{1-w^2}\right) = \frac{\sigma^6 w^2}{(1-w^2)^3}.$$

This proves that

$$\lim_{t\to\infty}\frac{1}{t^2}\mathbb{E}\left[\left(\sum_{i=1}^{t}x_i x_{i-1}\right)^2 x_t^2\right] = \frac{\sigma^6 w^2}{(1-w^2)^3},$$

as we aimed to show.

2) For the second equation, the proof is similar to that of the first equation. The expression whose limit we want to compute is given by

$$\frac{1}{t^2}\mathbb{E}\left[\left(\sum_{i=1}^{t}x_i^2\right)^2 x_t^2\right] = \frac{1}{t^2}\mathbb{E}\left[\sum_{i=1}^{t}x_i^4 x_t^2 + \sum_{j=2}^{t}x_i^2 x_j^2 x_t^2 \sum_{i=1}^{j-1}\right].$$

As in the previous case, only the double sum contributes to the limit $t\to\infty$. We compute, for $i < j$,

$$\mathbb{E}[x_i^2 x_j^2 x_t^2] = \mathbb{E}[x_i^2 x_j^2(w^{t-j}x_j + z_{j,t})^2]$$
$$= w^{2(t-j)}\mathbb{E}[x_i^2 x_j^4] + \mathbb{E}[x_i^2 x_j^2 z_{j,t}^2]$$
$$= w^{2(t-j)}\mathbb{E}[x_i^2 x_j^4] + \mathbb{E}[x_i^2(w^{j-i}x_i + z_{i,j})^2 z_{j,t}^2]$$
$$= w^{2(t-j)}\mathbb{E}[x_i^2 x_j^4] + w^{2(j-i)}\mathbb{E}[x_i^4 z_{j,t}^2] + \mathbb{E}[x_i^2 z_{i,j}^2 z_{j,t}^2].$$

Since $w^{2(t-j)}\mathbb{E}[x_i^2 x_j^4]$ and $w^{2(j-i)}\mathbb{E}[x_i^4 z_{j,t}^2]$ are $o(t^2)$, $\mathbb{E}[x_i^2 z_{i,j}^2 z_{j,t}^2]$ is the only term that contributes to the limit. We conclude that

$$\lim_{t\to\infty}\frac{1}{t^2}\mathbb{E}\left[\left(\sum_{i=1}^{t}x_i^2\right)^2 x_t^2\right] = \lim_{t\to\infty}\frac{2}{t^2}\sum_{j=2}^{t}\sum_{i=1}^{j-1}\mathbb{E}[x_i^2 z_{i,j}^2 z_{j,t}^2]$$
$$= \lim_{t\to\infty}\sum_{j=2}^{t}\sum_{i=1}^{j-1}\frac{\sigma^6(1-w^{2i})(1-w^{2(j-i)})(1-w^{2(t-j)})}{(1-w^2)^3}$$
$$= \frac{\sigma^6}{(1-w^2)^3}.$$

3) The proof of the third equation is similar to the proof of the first and second equations. We have

$$\mathbb{E}\left[\left(\sum_{i=1}^{t}x_i x_{i-1}\right)\left(\sum_{i=1}^{t}x_{i-1}^2\right)x_t^2\right] = \mathbb{E}\left[\sum_{i=1}^{t}x_i x_{i-1}^3 x_t^2 + 2\sum_{j=2}^{t}\sum_{i=1}^{j-1}x_j x_{j-1}x_{i-1}^2 x_t^2\right],$$

and only the double sum contributes to the limit. We compute that

$$
\begin{aligned}
\mathbb{E}[x_{i-1}^2 x_{j-1} x_j x_t^2] &= \mathbb{E}[x_{i-1}^2 x_{j-1} x_j (w^{t-j} x_j + z_{j,t})^2] \\
&= w^{2(t-j)} \mathbb{E}[x_{i-1}^2 x_{j-1} x_j^3] + \mathbb{E}[x_{i-1}^2 x_{j-1} x_j z_{j,t}^2] \\
&= w^{2(t-j)} \mathbb{E}[x_{i-1}^2 x_{j-1} x_j^3] + w \mathbb{E}[x_{i-1}^2 x_{j-1}^2 z_{j,t}^2] \\
&= w^{2(t-j)} \mathbb{E}[x_{i-1}^2 x_{j-1} x_j^3] + w \mathbb{E}[x_{i-1}^2 (w^{j-i} x_{i-1} + z_{i-1,j-1})^2 z_{j,t}^2] \\
&= w^{2(t-j)} \mathbb{E}[x_{i-1}^2 x_{j-1} x_j^3] + w^{2(j-i)+1} \mathbb{E}[x_{i-1}^4 z_{j,t}^2] + w \mathbb{E}[x_{i-1}^2 z_{i-1,j-1}^2 z_{j,t}^2].
\end{aligned}
$$

Since the first two terms are $o(t^2)$, only the third term contributes to the limit, and we conclude that

$$
\begin{aligned}
\lim_{t \to \infty} \frac{1}{t^2} \mathbb{E}\left[ \left( \sum_{i=1}^t x_i x_{i-1} \right) \left( \sum_{i=1}^t x_{i-1}^2 \right) x_t^2 \right] &= \lim_{t \to \infty} \frac{2w}{t^2} \sum_{j=2}^t \sum_{i=1}^t \mathbb{E}[x_{i-1}^2 z_{i-1,j-1}^2 z_{j,t}^2] \\
&= \lim_{t \to \infty} \frac{2w}{t^2} \sum_{j=2}^t \sum_{i=1}^t \frac{\sigma^6 (1 - w^{2(i-1)})(1 - w^2(j-i))(1 - w^{2(t-j)})}{(1 - w^2)^3} \\
&= \frac{\sigma^6 w}{(1 - w^2)^3}.
\end{aligned}
$$

$\square$

The following lemma controls the size $z_{T,L}$ along the Richardson iteration and is used in the proof of Theorem 1.

**Lemma 11.** *Let $(x_0, \ldots, x_t)$ follow the dynamical system (2) and let $z_{T,L}$ be defined by the Richardson iteration*

$$
\begin{cases}
z_{T,L} = z_{T,L-1} + \alpha \left( x_T - X_T z_{T,L-1} \right), \ L > 0 \\
z_{T,0} = 0.
\end{cases}
$$

*Then, as long as $\alpha < \frac{2(1 - w_{\max}^2)}{c\sigma^2}$ for a numerical constant $c > 0$, we have*

$$
\mathbb{E}[\|z_{T,L}\|^k] = O\left( k \left( 1 + L^{2k} \alpha^{2k} \right) + \alpha^{2k(L-1)} T^{-k(L-1)} \cdot (2k(L-1))! \right).
$$

*Proof.* By Fubini's theorem, we have

$$
\mathbb{E}[\|z_{T,L}\|^k] = \int_0^\infty \mathbb{P}\left( \|z_{T,L}\|^k > r \right) dr.
$$

Changing variables, we find

$$
\begin{aligned}
\mathbb{E}[\|z_{T,L}\|^k] &= \int_0^\infty k r^{k-1} \mathbb{P}\left( \|z_{T,L}\| > r \right) dr \\
&\leq k + \int_1^\infty k r^{k-1} \mathbb{P}\left( \|z_{T,L}\| > r \right) dr.
\end{aligned}
$$

By the recurrence defining $z_{T,L}$ and the triangle inequality, we have

$$
\mathbb{P}\left( \|z_{T,L}\| > r \right) \leq \mathbb{P}\left( \|(\mathbf{I}_d - \alpha X_T)\|_{\mathrm{op}} \|z_{T,L-1}\| + \alpha \|x_T\| > r \right).
$$

Iterating this bound, we have

$$
\int_1^\infty k r^{k-1} \mathbb{P}\left( \|z_{T,L}\| > r \right) dr \leq \int_1^\infty k r^{k-1} \mathbb{P}\left( \alpha \sum_{j=1}^{L-1} \|\mathbf{I}_d - \alpha X_T\|_{\mathrm{op}}^j \|x_T\| > r \right) dr.
$$

If $\mathcal{E}$ denotes the event on which $\|(\mathbf{I}_d - \alpha X_T)\| \le 1$, then we have

$$\int_1^\infty kr^{k-1} \mathbb{P}\left(\alpha \sum_{j=1}^{L-1} \|(\mathbf{I}_d - \alpha X_T)\|_{\mathrm{op}}^j \|x_T\| > r\right) dr$$

$$= \int_1^\infty kr^{k-1} \mathbb{P}\left(\alpha \sum_{j=1}^{L-1} \|(\mathbf{I}_d - \alpha X_T)\|_{\mathrm{op}}^j \|x_T\| > r \cap \mathcal{E}\right) dr$$

$$+ \int_1^\infty kr^{k-1} \mathbb{P}\left(\alpha \sum_{j=1}^{L-1} \|(\mathbf{I}_d - \alpha X_T)\|_{\mathrm{op}}^j \|x_T\| > r \cap \mathcal{E}^c\right) dr$$

For the first term, notice that

$$\int_1^\infty kr^{k-1} \mathbb{P}\left(\alpha \sum_{j=1}^{L-1} \|(\mathbf{I}_d - \alpha X_T)\|_{\mathrm{op}}^j \|x_T\| > r \cap \mathcal{E}\right) \le \int_1^\infty kr^{k-1} \mathbb{P}\left(\alpha L \|x_t\| > r\right) dr$$

$$\le (\alpha L)^k \int_0^\infty kr^{k-1} \mathbb{P}\left(\|x_T\|\| > r\right) dr$$

$$= O\left(kL^k \alpha^k\right),$$

where we note that the above integral converges due to the exponential concentration of $\|x_T\|$ proven in Lemma 12. For the second term, since $\|\mathbf{I}_d - \alpha X_t\|_{\mathrm{op}} > 1$ on $\mathcal{E}^c$, we have

$$\int_1^\infty kr^{k-1} \mathbb{P}\left(\alpha \sum_{j=1}^{L-1} \|I - \alpha X_T\|_{\mathrm{op}}^j \|x_T\| > r \cap \mathcal{E}^c\right) dr \le \int_1^\infty kr^{k-1} \mathbb{P}\left(L\alpha \|\mathbf{I}_d - \alpha X_T\|_{\mathrm{op}}^{L-1} \|x_T\| > r\right) dr$$

$$= \int_1^\infty kr^{k-1} \mathbb{P}\left(\alpha L \|I - \alpha X_T\|_{\mathrm{op}}^{L-1} \|x_T\| > r \cap \{\|x_T\| \le \sqrt{r}\}\right) dr$$

$$+ \int_1^\infty kr^{k-1} \mathbb{P}\left(\alpha L \|I - \alpha X_T\|_{\mathrm{op}}^{L-1} \|x_T\| > r \cap \{\|x_T\| > \sqrt{r}\}\right) dr$$

$$\le \int_1^\infty kr^{k-1} \mathbb{P}\left(\alpha L \|I - \alpha X_T\|_{\mathrm{op}}^{L-1} > \sqrt{r}\right) dr + \int_1^\infty kr^{k-1} \mathbb{P}\left(\|x_T\| > \sqrt{r}\right) dr.$$

Notice that by the concentration bound for $\|x_T\|^2$ stated in Lemma 12, the second term above is bounded by

$$\int_1^\infty kr^{k-1} \mathbb{P}\left(\|x_T\| > \sqrt{r}\right) dr \le C_1 \int_1^\infty kr^{k-1} \exp\left(-\frac{r}{C_2}\right) dr = O(k)$$

where $C_1$ and $C_2$ are defined in Lemma 12. For the first term, we make the change of variables $r \mapsto \left(\frac{r^{1/2}}{\alpha L}\right)^{1/(L-1)}$ to rewrite the integral as

$$\int_1^\infty kr^{k-1} \mathbb{P}\left(\alpha L \|\mathbf{I}_d - \alpha X_T\|_{\mathrm{op}}^{L-1} > \sqrt{r}\right) dr$$

$$= k(\alpha L)^{2k} \int_{(\alpha L)^{1/(L-1)}}^\infty r^{2k(L-1)-1} \mathbb{P}\left(\|(\mathbf{I}_d - \alpha X_T)\|_{\mathrm{op}} > r\right) dr.$$

We further have

$$\int_{(\alpha L)^{1/(L-1)}}^\infty r^{2k(L-1)-1} \mathbb{P}\left(\|(\mathbf{I}_d - \alpha X_T)\|_{\mathrm{op}} > r\right) dr \le \int_{(\alpha L)^{1/(L-1)}}^1 r^{2k(L-1)-1} \mathbb{P}\left(\|(\mathbf{I}_d - \alpha X_T)\|_{\mathrm{op}} > r\right) dr$$

$$+ \int_1^\infty r^{2k(L-1)-1} \mathbb{P}\left(\|(\mathbf{I}_d - \alpha X_T)\|_{\mathrm{op}} > r\right) dr$$

$$\le 1 + \int_1^\infty r^{2k(L-1)-1} \mathbb{P}\left(\|(\mathbf{I}_d - \alpha X_T)\|_{\mathrm{op}} > r\right) dr.$$

Note that the first integral in the sum above could be negative if $\alpha L > 1$. This proves that

$$k\,(\alpha L)^{2k} \int_{(\alpha L)^{1/(L-1)}}^{\infty} r^{2k(L-1)-1} \mathbb{P}\left(\|(\mathbf{I}_d - \alpha X_T)\|_{\mathrm{op}} > r\right) dr$$

$$\leq k(\alpha L)^k + \int_1^{\infty} r^{2k(L-1)-1} \mathbb{P}\left(\|(\mathbf{I}_d - \alpha X_T)\|_{\mathrm{op}} > r\right) dr.$$

To control the remaining integral, we note that

$$\|(I - \alpha X_T)\|_{\mathrm{op}} = \max\left(|1 - \alpha\lambda_{\min}(X_T)|, |1 - \alpha\lambda_{\max}(X_T)|\right)$$
$$\leq \max\left(1, \alpha\lambda_{\max}(X_T) - 1\right).$$

Therefore, for $r > 1$,

$$\mathbb{P}\left(\|(I - \alpha X_T)\|_{\mathrm{op}} > r\right) \leq \mathbb{P}\left(\left(\lambda_{\max}(X_T) > \frac{1+r}{\alpha}\right)\right).$$

This allows us to bound the remaining integral by

$$\int_1^{\infty} r^{2k(L-1)-1} \mathbb{P}\left(\|\mathbf{I}_d - \alpha X_T\|_{\mathrm{op}} > r\right) dr \leq \int_1^{\infty} r^{2k(L-1)-1} \mathbb{P}\left(\lambda_{\max}(X_T) > \frac{r+1}{\alpha}\right) dr$$

$$= \alpha \int_{2/\alpha}^{\infty} (\alpha r - 1)^{2k(L-1)-1} \mathbb{P}\left(\lambda_{\max}(X_T) > r\right) dr$$

$$\leq \alpha^{2k(L-1)} \int_{2/\alpha}^{\infty} r^{2k(L-1)-1} \mathbb{P}\left(\lambda_{\max}(X_T) > r\right) dr$$

$$\leq \alpha^{2k(L-1)} \int_{2/\alpha}^{\infty} r^{2k(L-1)-1} \exp\left(-\frac{c\sqrt{T}r(1 - w_{\max}^2)}{\sigma^4} + \frac{c\sqrt{T}}{\sigma^2}\right) dr,$$

where we used the concentration inequality proved in Lemma 12 in the last line. After another change of variables, we can rewrite the above integral as

$$\alpha^{2k(L-1)} \int_{2/\alpha}^{\infty} r^{2k(L-1)-1} \exp\left(-\frac{c\sqrt{T}r(1 - w_{\max}^2)}{\sigma^4} + \frac{c\sqrt{T}}{\sigma^2}\right) dr$$

$$= \alpha^{2k(L-1)} \left(\frac{\alpha\sigma^4}{c\sqrt{T}(1 - w_{\max}^2)}\right)^{2k(L-1)} \int_{\frac{\sigma^4}{c\sqrt{T}r(1 - w_{\max}^2)}\left(\frac{2}{\alpha} - \frac{c\sigma^2}{1 - w_{\max}^2}\right)}^{\infty} r^{2k(L-1)-1} e^{-r} dr.$$

If $\alpha < \frac{2(1 - w_{\max}^2)}{c\sigma^2}$, then the lower limit of the above integral is non-negative, and hence we have the upper bound

$$\alpha^{2k(L-1)} \left(\frac{\alpha\sigma^4}{c\sqrt{T}(1 - w_{\max}^2)}\right)^{2k(L-1)} \int_{\frac{\sigma^4}{c\sqrt{T}r(1 - w_{\max}^2)}\left(\frac{2}{\alpha} - \frac{c\sigma^2}{1 - w_{\max}^2}\right)}^{\infty} r^{2k(L-1)-1} e^{-r} dr$$

$$\leq \alpha^{2k(L-1)} \left(\frac{\alpha\sigma^4}{c\sqrt{T}(1 - w_{\max}^2)}\right)^{2k(L-1)} \Gamma(2k(L-1))$$

$$= \alpha^{2k(L-1)} \left(\frac{\alpha\sigma^4}{c\sqrt{T}(1 - w_{\max}^2)}\right)^{2k(L-1)} \cdot (2k(L-1))!$$

where $\Gamma(\cdot)$ denotes the $\Gamma$ function. Combining the estimates for each term gives us the bound as stated in the Lemma. $\qquad\square$

We make use of the following elementary concentration inequalities for the dynamical system.

**Lemma 12.** *Let $(x_0, \ldots, x_T)$ be defined by the dynamical system* (2). *Then*

*1.* $\mathbb{P}\left(\|x_T\| \geq r\right) \leq \exp\left(\frac{d(1 - w_{\max}^2)^2}{8\sigma^2}\right) \cdot \exp\left(-\frac{r(1 - w_{\max}^2)^2}{8\sigma^4}\right).$

2. $\mathbb{P}\left(\lambda_{\max}\left(\frac{1}{T}\sum_{i=1}^{T}x_ix_i^T\right)>r\right)\le\exp\left(-\frac{c\sqrt{T}r(1-w_{\max}^2)}{\sigma^4}+\frac{c\sqrt{T}}{\sigma^2}\right).$

*Proof.* To prove 1), note that by Lemma 8, $x_T$ is a centered Gaussian random vector with covariance

$$\operatorname{cov}(x_T)=\sigma^2(\mathbf{I}_d-W^{2T})(\mathbf{I}_d-W^2)^{-1}\prec\frac{\sigma^2}{1-w_{\max}^2}\cdot\mathbf{I}_d.$$

It follows that $\|x_T\|^2$ can be written as a sum of $d$ independent random variables

$$\|x_T\|^2=\sum_{j=1}^{d}x_{T,j}^2,$$

where $x_{T,j}$ are normal with $\mathbb{E}[x_{T,j}]=0$ and $\mathbb{E}[x_{T,j}^2]\le\frac{\sigma^2}{1-w_{\max}^2}$. In particular, each $x_{T,j}$ is sub-exponential with parameters $\left(2,4\left(\frac{\sigma^2}{1-w_{\max}^2}\right)^2\right)$ (see Wainwright [2019]). This implies that $\|x_T\|^2$ is sub-exponential with parameters $\left(2\sqrt{d},4\left(\frac{\sigma^2}{1-w_{\max}^2}\right)^2\right)$, which yields the concentration inequality

$$\mathbb{P}\left(\|x_T\|^2>r\right)=\mathbb{P}\left(\|x_T\|^2-\mathbb{E}[\|x_T\|^2]>r-\mathbb{E}[\|x_T\|^2]\right)$$
$$\le\exp\left(-\frac{(r-\mathbb{E}[\|x_T\|^2])(1-w_{\max}^2)^2}{8\sigma^4}\right)$$
$$\le\exp\left(\frac{d(1-w_{\max}^2)^2}{8\sigma^2}\right)\cdot\exp\left(-\frac{r(1-w_{\max}^2)^2}{8\sigma^4}\right).$$

To prove 2), we proceed in two steps.

**Step 1:** we first show that the 1D dynamical system
$$z_{t+1}=wz_t+\xi_{t+1},\ \xi_{t+1}\sim N(0,\sigma^2),\ w\in[w_{\min},w_{\max}],\ z_0=0$$
satisfies the concentration inequality

$$\mathbb{P}\left(\frac{1}{T}\sum_{i=1}^{T}z_i^2>r\right)\le\exp\left(\frac{d(1-w_{\max}^2)^2}{8\sigma^2}\right)\cdot\exp\left(-\frac{r(1-w_{\max}^2)^2}{8\sigma^4}\right).$$

To prove this, note that $z_i=\sum_{j=1}^{i}w^{i-j}\xi_j$, and therefore

$$\frac{1}{T}\sum_{i=1}^{T}z_i^2=\frac{1}{T}\sum_{i=1}^{T}\sum_{j,k=1}^{T}w^{2i-(j+k)}\xi_j\xi_j=\xi^TA\xi,$$

where $\xi=(\xi_1,\ldots,\xi_T)\in\mathbb{R}^T$ and $A\in\mathbb{R}^{T\times T}$ is defined by

$$A_{jk}=\frac{1}{T}\sum_{i=\max(j,k)}^{T}w^{2i-(j+k)}$$
$$=\frac{1}{T}\left(w^{2\max(j,k)-(j+k)}+\cdots+w^{2T-(j+k)}\right)$$
$$\le\frac{1}{T}\cdot w^{2\max(j,k)-(j+k)}\cdot\frac{1}{1-w^2}.$$

It follows that

$$\|A\|_F^2\le\frac{1}{T^2(1-w^2)^2}\sum_{j,k=1}^{T}w^{4\max(j,k)-2(j+k)}$$
$$=\frac{2}{T^2(1-w^2)^2}\sum_{k=1}^{T}\sum_{j=1}^{k}w^{2(k-j)}$$
$$=\frac{2}{T^2(1-w^2)^2}\sum_{k=1}^{T}\frac{1-w^{2k}}{1-w^2}$$
$$\le\frac{2}{T(1-w^2)^2}.$$

Hence

$$\|A\|_F \le \frac{\sqrt{2}}{(1-w^2)\sqrt{T}}.$$

Then, with $\xi = (\xi_1, \ldots, \xi_t)$, noting that each $\xi_i$ is $\sigma^2$-sub-Gaussian, the Hanson-Wright inequality (Vershynin [2018]) guarantees that

$$\mathbb{P}\left(\xi^T A \xi - \mathbb{E}[\xi^T A \xi] > r\right) \le \exp\left(-\frac{c\sqrt{T}r(1-w_{\max}^2)}{\sigma^4}\right),$$

where $c > 0$ is a universal constant. Noting that $\mathbb{E}[\xi^T A \xi] \le \frac{\sigma^2}{1-w_{\max}^2}$ by Lemma 8 and rearranging the bound above, we find that

$$\mathbb{P}\left(\xi^T A \xi > r-\right) \le \exp\left(\frac{c\sqrt{T}(r - \mathbb{E}[\xi^T A \xi])(1-w_{\max}^2)}{\sigma^4}\right)$$

$$\le \exp\left(-\frac{c\sqrt{T}r(1-w_{\max}^2)}{\sigma^4} + \frac{c\sqrt{T}}{\sigma^2}\right).$$

**Step 2:** We show that the bound above implies the desired concentration bound for $\lambda_{\max}\left(\frac{1}{T}\sum_{i=1}^T x_i x_i^T\right)$. To begin, note that

$$\mathbb{P}\left(\lambda_{\max}\left(\frac{1}{T}\sum_{i=1}^T x_i x_i^T\right) > r\right) \le \mathbb{P}\left(\frac{1}{T}\sum_{i=1}^T \|x_i\|^2 > r\right).$$

Next, by expanding each norm $\|x_i\|^2$ in the basis of eigenvectors of $W$, we can write

$$\frac{1}{T}\sum_{i=1}^T \|x_i\|^2 = \frac{1}{T}\sum_{i=1}^T \sum_{j=1}^d z_{i,j}^2,$$

where each $(z_{1,j}, \ldots, z_{T,j})$ is an independent copy of the 1D dynamical system defined in Step 1. By the Bernstein bound for sums of subexponential random variables, this implies that

$$\mathbb{P}\left(\frac{1}{T}\sum_{i=1}^T \|x_i\|^2 > r\right) \le \exp\left(-\frac{c\sqrt{T}r(1-w_{\max}^2)}{\sigma^4} + \frac{c\sqrt{T}}{\sigma^2}\right),$$

and therefore that

$$\mathbb{P}\left(\lambda_{\max}\left(\frac{1}{T}\sum_{i=1}^T x_i x_i^T\right) > r\right) \le \exp\left(-\frac{c\sqrt{T}r(1-w_{\max}^2)}{\sigma^4} + \frac{c\sqrt{T}}{\sigma^2}\right).$$

$\square$

The following lemma is used to extend pointwise convergence of functions to uniform convergence. Recall that a sequence of functions $\{f_n\}$ on a compact metric space $\mathcal{X}$ is *equicontinuous* at $x \in \mathcal{X}$ if for every $\epsilon > 0$, there exists a $\delta > 0$ such that for all $x' \in \mathcal{X}$ with $|x - x'| < \delta$, we have $\sup_n |f_n(x) - f_n(x')| < \epsilon$. The sequence $\{f_n\}$ is *equicontinuous* if it is equicontinuous at every $x \in \mathcal{X}$.

**Lemma 13.** *Let $\{f_n\}$, $f$ be functions on a compact metric space $\mathcal{X}$ such that*

    *1. The sequence $\{f_n\}$ is equicontinuous,*

    *2. $f_n \to f$ pointwise.*

*Then $f_n \to f$ uniformly.*

*Proof.* Since $f_n \to f$ pointwise, the sequence $\{f_n(x)\}$ is bounded for each $x \in \mathcal{X}$. Therefore the sequence of functions $\{f_n\}$ is pointwise bounded and equicontinuous, so by the Arzela-Ascoli Theorem (see e.g. Folland [1999]), it has a uniformly convergent subsequence. But the limit of any uniformly convergent subsequence of $\{f_n\}$ must equal the pointwise limit of the sequence. This proves that $f$ is the only subsequential limit of $\{f_n\}$ in the uniform topology, and hence $f_n \to f$ uniformly. $\square$

