# OpenReview forum: "In-context Learning of Linear Dynamical Systems with Transformers: Approximation Bounds and Depth-separation"
_NeurIPS.cc/2025/Conference — NeurIPS 2025 poster_

### Official Review · Reviewer_ETpH · 2025-06-22

**Clarity:** 4
**Significance:** 4
**Originality:** 4
**Rating:** 5
**Confidence:** 3

**Summary:**

This paper studies the in-context learning capabilities of linear transformers. The authors propose the following problem: learning, using a model parametrized by a multi-layer linear transformer, on sequences of next-token prediction trajectory that take the form of noisy linear dynamical systems, with context length equal to one, and design matrices of each LDS being sampled according to some distribution.
After presenting this problem, the results of the paper are two-folds. First, the authors prove that provided the depth is larger than logarithmic in $T$ (the length of each trajectory), there exists a linear transformer that predicts well the last iterate of such LDSs (i.e., performs in context learning of these tasks), with a test error that scales as $\log(T)/T$. Then, the authors prove that depth 1 is not enough to have a test error that decreases to 0.

**Questions:**

I have several questions and remark:
Q1) How about long-context problems such as long-context linear dynamical systems (such as e.g. *Long-Context Linear System Identification, Yuksel et al., ICLR 2025*)?
Q2) What would be the authors take on the optimization perspectives of the problem?
Q3) In Eq. (4), I agree that putting a $sup_W$ in the test loss captures a form of worst-case robustness. However, would the depth separation result appear also if it were an average over the sampled matrices ?

**Ethical Concerns:**

["NO or VERY MINOR ethics concerns only"]

**Quality:**

4

**Strengths And Weaknesses:**

**Strengths**
S1) In-context learning capacities of transformers is a very timely subject, and the simplified problem studied in this paper is a very nice and elegant proxy to study this problem. It of course does not capture the complexity of real LLMs, but that's completely out of reach for now.
S2) The depth-separation result is quite strong in my opinion: theory of deep learning and LLMs hardly capture such results, as far as I know. Such a result in the context of next-token prediction and ICL really is a strength.
S3) The paper is overall very well written: intuitions, assumptions, results are clearly stated, in a way that is reader-friendly. I did not read the proofs fully, but skimming through them and the help of the proof sketch, makes them overall clear.


**Weaknesses**
W1) The proposed results characterize the fact that if a certain depth is satisfied, there exists some linear transformer that satifies desired properties. However, this does not takes into account optimization perspectives of the problem.


Overall, I think the contributions and the clarity of this submission are quite strong, and definitely make it above the acceptance threshold.

---

> ### Author Rebuttal · Authors · 2025-07-31
>
> We thank the reviewer for their kind and helpful feedback and appreciate the positive assessment of our work. First, since several reviewers cited similar weaknesses, we have a general response:
>
> $\\textbf{Experiments:}$ We ran some experiments to validate our theoretical results. Due to the current rebuttal guidelines, we cannot show the plots from our experiments. However, we plan to add our experimental results to the revised version. Our results are as follows. We trained transformers with both linear and softmax attention to learn random linear dynamical systems in-context. For computational reasons, the loss we use is averaged over the task distribution, as opposed to the supremum loss we use for theory. For one-dimensional systems with noise variance $\sigma^2 = 0.05$ and context length $T = 500$, we present the minimum test loss as as a function of depth in the following tables:
>
> $$
> \\begin{bmatrix}
> \\textrm{Number of layers (softmax attention)} & \\textrm{Minimum loss value:} \\\\
> L=1 & 0.0011824991088360548 \\\\
> L=2 & 0.00022110833378974348 \\\\
> L=5 & 0.00012664416863117367
> \\end{bmatrix}
> $$
>
> $$
> \\begin{bmatrix}
> \\textrm{Number of layers (linear attention)} & \\textrm{Minimum loss value:} \\\\
> L=1 & 0.0007882040226832032 \\\\
> L=2 & 0.00015281644300557673 \\\\
> L=5 & 0.00011635164264589548
> \\end{bmatrix}
> $$
>
> We also have numerical experiments for multi-dimensional systems (up to $d=5$), but the results are most illustrative for one-dimensional examples. Additionally, while we are unable to produce results for depth greater than $L=5$ due to the time constraint, we plan to include experimental results for deeper transformers in the revised version. Nonetheless, we hope that our inclusion of these experimental results strengthens the overall contribution of the paper. To summarize, our plots confirm the following points:
>
> 1. There is a clear separation in the error achieved by single-layer and multi-layer transformers. This provides evidence for Theorem 2 in our paper.
> 2. Increasing the depth beyond two layers  marginally improves the error, but the most significant difference is between one and two-layer transformers.
> 3. For each choice of the number of layers $L$, the minimum loss achieved by the softmax attention transformer is comparable to the minimum loss achieved by the linear attention transformer. This suggests that the non-vanishing lower bound persists when using softmax attention. The minimum loss values are shown in the table below.
>
> $\\textbf{Choice of architecture for the lower bound:}$ Broadly, the goal of our paper is to provably demonstrate the importance of depth for in-context learning when the data is not iid. We think that the linear transformer is a natural architecture through which to study this phenomenon, since several prior works have studied the in-context learning ability of  this architecture on iid data. In particular, the recent works [1,2,3] which prove upper bounds on the loss for single-layer linear transformers on iid data, provide a direct baseline to compare with our lower bound. Currently, we are investigating  whether our upper bounds can be extended to linear transformers with finite depth, as well as the approximation error of nonlinear transformers, such as those with with softmax attention modules; however, these directions are beyond the scope of the current work.
>
> Concerning your specific questions:
>
> 1. (Extension to long-context problems): It would be very interesting to extend the positive results of our paper to linear dynamical systems which depend on a context length $p > 1$. Since our proof for the case $p=1$ constructs a transformer which implements the least-squares estimator, the statistical guarantees derived in [4] of constrained least-squares estimators could potentially be used to adapt the proof of our result to the long-context case (although the transformer construction would have to be modified, since the current proof uses the specific form of the least-squares predictor in the $p=1$ case). We plan to add a discussion of this problem in the conclusion of the paper and cite the reference [4].
>
> 2. (Optimization dynamics): In general, the optimization dynamics of multi-layer transformers for this learning task is highly nontrivial and beyond the scope of this work. However, recent work [5] has demonstrated that transformers linear transformers can provably learn to perform multi-step gradient descent for in-context linear regression using chain-of-thought (CoT) training. Since the ICL mechanism in [5] is similar to ours (multi-step gradient descent vs multi-step Richardson iteration), a natural extension of our work is to study the dynamics of CoT training to enforce the construction of Theorem 1. We plan to investigate this direction more carefully in the future.
>
> 3. ($\sup$ loss versus average loss): The reviewer asks an interesting question. Our experiments, which use the average loss instead of sup loss for computational reasons, suggest that a depth-separation is still present in this case. The heuristic behind the lower bound is  that the covariance of a trajectory depends on the task $W$, and thus the expressions in the least-squares predictor which involve the covariance cannot simply be implemented directly by the attention weights. The $\sup$ loss naturally captures this limitation, but we plan to investigate whether a non-vanishing lower bound also holds under the $W$-averaged loss in the future.
>
> References:
>
> [1] Zhang, Ruiqi, Spencer Frei, and Peter L. Bartlett. "Trained transformers learn linear models in-context." Journal of Machine Learning Research 25.49 (2024): 1-55.
>
> [2] Ahn, Kwangjun, et al. "Transformers learn to implement preconditioned gradient descent for in-context learning." Advances in Neural Information Processing Systems 36 (2023): 45614-45650.
>
> [3] Mahankali, Arvind, Tatsunori B. Hashimoto, and Tengyu Ma. "One step of gradient descent is provably the optimal in-context learner with one layer of linear self-attention." arXiv preprint arXiv:2307.03576 (2023)
>
> [4] Yüksel, Oğuz Kaan, Mathieu Even, and Nicolas Flammarion. "Long-Context Linear System Identification." arXiv preprint arXiv:2410.05690 (2024).
>
> [5] Huang, Jianhao, Zixuan Wang, and Jason D. Lee. "Transformers learn to implement multi-step gradient descent with chain of thought." arXiv preprint arXiv:2502.21212 (2025).

---

> > ### Comment · Reviewer_ETpH · 2025-08-01
> > **Answer to the rebuttal**
> >
> > Dear authors,
> >
> > Thanks a lot for the detailed answer.
> >
> > I had not raised any questions about experiments, but to me *There is a clear separation in the error achieved by single-layer and multi-layer transformers. This provides evidence for Theorem 2 in our paper* is not that direct, in light also of the optimization perspective highighted in the review and answered in the rebuttal. Indeed, what this experiment shows is mostly a motivation for studying the same problem, but with an optimization perspective. I agree with the authors that this would be a much harder problem, completely out of scope for this paper. I however disagree with the conclusions from this experiment.
> > Based on *Since the ICL mechanism in [5] is similar to ours (multi-step gradient descent vs multi-step Richardson iteration), a natural extension of our work is to study the dynamics of CoT training to enforce the construction of Theorem 1.* in the rebuttal, I think this direction is very promising.
> >
> > I thank the authors for answers to 1. and 3., that are satisfying.
> > I will keep my score as is.

---

> > > ### Author Response · Authors · 2025-08-01
> > >
> > > Thank you again for your support of the contributions and for your valuable feedback. We sincerely appreciate the work you have done to review our paper.

---

### Official Review · Reviewer_22f5 · 2025-06-30

**Clarity:** 2
**Significance:** 2
**Originality:** 3
**Rating:** 4
**Confidence:** 4

**Summary:**

This paper theoretically studies ICL in the non-IID setting, within the linear dynamica system frameword. A upper bound for error of a multi-layer linear transformer and the error lower bounded for a singla layer linear transformer is provided.

**Questions:**

See Weakness section. Besides, is it possible to extend the lower bound to a multi-head transformer with linear attention?

**Ethical Concerns:**

["NO or VERY MINOR ethics concerns only"]

**Final Justification:**

The rebuttal has resolved most of my initial concerns, and thus I will maintain my score.

**Limitations:**

yes

**Quality:**

3

**Strengths And Weaknesses:**

Strengths:

1). This work provides a theoretical study of the ICL capabilities of transformers for linear dynamical systems, which is a novel contribution.

2). The lower bound result demonstrating that a single-layer linear transformer lacks sufficient expressive power for in-context learning of linear dynamical systems, to me, it is particularly interesting.

Weakness:

1). Some existing works have already studied ICL for non-IID data theoretically, even if not explicitly framed as such. For example, "Non-asymptotic Convergence of Training Transformers for Next-Token Prediction".

2). Theorem 1 focuses on the expressiveness of transformers, showing that they can efficiently optimize training loss for linear dynamical systems by mimicking an iterative algorithm via construction. However, this result does not guarantee that a transformer will actually converge to such a solution during pre-training.

3). Theorems 1 and 2 rely on a linear transformer architecture, which is quite restrictive. While this is not a major criticism, introducing non-linear activations in the attention block would strengthen the theoretical contribution, especially for the lower bound.

4). The lower bound result is based on constrained attention and mlp layers. Although the authors discuss a few sentences about the fully parameterized case, no theoretical results are provided. Could the constraints on the linear attention blocks be relaxed, even if the constraints for mlp layer are kept?

5).  Experimental validation of Theorem 2 would strengthen the paper, since it would show the incapability of a linear transformer structure with one layer.

---

> ### Author Rebuttal · Authors · 2025-07-31
>
> We thank the reviewer for their kind and helpful feedback. Since several reviewers cited the same weaknesses, we have the following general response:
>
> $\\textbf{Experiments:}$ We ran some experiments to validate our theoretical results. Due to the current rebuttal guidelines, we cannot show the plots from our experiments. However, we plan to add our experimental results to the revised version. Our results are as follows. We trained transformers with both linear and softmax attention to learn random linear dynamical systems in-context. For computational reasons, the loss we use is averaged over the task distribution, as opposed to the supremum loss we use for theory. For one-dimensional systems with noise variance $\sigma^2 = 0.05$ and context length $T = 500$, we present the minimum test loss as as a function of depth in the following tables:
>
> $$
> \\begin{bmatrix}
> \\textrm{Number of layers (softmax attention)} & \\textrm{Minimum loss value:} \\\\
> L=1 & 0.0011824991088360548 \\\\
> L=2 & 0.00022110833378974348 \\\\
> L=5 & 0.00012664416863117367
> \\end{bmatrix}
> $$
>
> $$
> \\begin{bmatrix}
> \\textrm{Number of layers (linear attention)} & \\textrm{Minimum loss value:} \\\\
> L=1 & 0.0007882040226832032 \\\\
> L=2 & 0.00015281644300557673 \\\\
> L=5 & 0.00011635164264589548
> \\end{bmatrix}
> $$
>
> We also have numerical experiments for multi-dimensional systems (up to $d=5$), but the results are most illustrative for one-dimensional examples. Additionally, while we are unable to produce results for depth greater than $L=5$ due to the time constraint, we plan to include experimental results for deeper transformers in the revised version. Nonetheless, we hope that our inclusion of these experimental results strengthens the overall contribution of the paper. To summarize, our plots confirm the following points:
>
> 1. There is a clear separation in the error achieved by single-layer and multi-layer transformers. This provides evidence for Theorem 2 in our paper.
> 2. Increasing the depth beyond two layers  marginally improves the error, but the most significant difference is between one and two-layer transformers.
> 3. For each choice of the number of layers $L$, the minimum loss achieved by the softmax attention transformer is comparable to the minimum loss achieved by the linear attention transformer. This suggests that the non-vanishing lower bound persists when using softmax attention. The minimum loss values are shown in the table below.
>
> $\\textbf{Choice of architecture for the lower bound:}$ Broadly, the goal of our paper is to provably demonstrate the importance of depth for in-context learning when the data is not iid. We think that the linear transformer is a natural architecture through which to study this phenomenon, since several prior works have studied the in-context learning ability of  this architecture on iid data. In particular, the recent works [1,2,3] which prove upper bounds on the loss for single-layer linear transformers on iid data, provide a direct baseline to compare with our lower bound. Currently, we are investigating  whether our upper bounds can be extended to linear transformers with finite depth, as well as the approximation error of nonlinear transformers, such as those with with softmax attention modules; however, these directions are beyond the scope of the current work.
>
> Concerning your specific questions:
>
> 1. (Existing works): We had not encountered the work [4], but we agree that it also studies transformers' ability to learn from non-IID data. Compared to our work, the setting is quite different, since [4] focuses on sentences generated from a discrete vocabulary, whereas we study linear dynamical systems on the continuous state space $\mathbb{R}^d$. We have updated the literature review of our paper to include a discussion of [4] and other relevant papers  we missed.
>
> 2. (Optimization): In general, the optimization dynamics of multi-layer transformers for this learning task is highly nontrivial and beyond the scope of this work. However, recent work [5] has demonstrated that transformers linear transformers can provably learn to perform multi-step gradient descent for in-context linear regression using chain-of-thought (CoT) training. Since the ICL mechanism in [5] is similar to ours (multi-step gradient descent vs multi-step Richardson iteration), a natural extension of our work is to study the dynamics of CoT training to enforce the construction of Theorem 1. We leave the investigation of this direction to future work.
>
> 3. (Using full attention parameterization and multi-head attention): We suspect the same proof strategy can be used to prove a lower bound for the full parameterization and for multi-head attention, as it just amounts to more additive terms in the limiting loss function. For the single-head case, we argue that the parameters we chose to zero out are not necessary to predict the dynamical system. Indeed, we know that the next token $x_{t+1}$ takes the form $x_{t+1} = Wx_t + \textrm{noise}$, and the loss function minimizes the difference between the transformer prediction $y_{\theta}$ and the conditional mean of $x_{t+1}$ given the history $x_1, \dots, x_t$. This motivates us to parameterize the transformer to output an estimator of the form $y_{\theta} =W_{\theta} x_t,$ where $W_{\theta}$ is a matrix depending on the transformer parameters $\theta$ and the history $x_1, \dots, x_t$. Under the full parameterization, the transformer output is of the form $y_{\theta} =W_{\theta}^{(1)} x_t +W_{\theta}^{(2)} x_{t-1}.$ This motivates us to choose a parameterization of the transformer which corresponds to setting $W_{\theta}^{(2)} = 0$.
>
> References:
>
> [1] Zhang, Ruiqi, Spencer Frei, and Peter L. Bartlett. "Trained transformers learn linear models in-context." Journal of Machine Learning Research 25.49 (2024): 1-55.
>
> [2] Ahn, Kwangjun, et al. "Transformers learn to implement preconditioned gradient descent for in-context learning." Advances in Neural Information Processing Systems 36 (2023): 45614-45650.
>
> [3] Mahankali, Arvind, Tatsunori B. Hashimoto, and Tengyu Ma. "One step of gradient descent is provably the optimal in-context learner with one layer of linear self-attention." arXiv preprint arXiv:2307.03576 (2023)
>
> [4] Huang, Ruiquan, Yingbin Liang, and Jing Yang. "Non-asymptotic convergence of training transformers for next-token prediction." Advances in Neural Information Processing Systems 37 (2024): 80634-80673.
>
> [5] Huang, Jianhao, Zixuan Wang, and Jason D. Lee. "Transformers learn to implement multi-step gradient descent with chain of thought." arXiv preprint arXiv:2502.21212 (2025).

---

### Official Review · Reviewer_Q9Jh · 2025-07-02

**Clarity:** 4
**Significance:** 3
**Originality:** 2
**Rating:** 4
**Confidence:** 3

**Summary:**

The paper studies the in-context learning of linear dynamical systems for transformers and presents two main results. The first is the approximation error for multi-layer transformers which implements the Richardson iteration. The second is a lower bound for a single-layer transformer.

**Questions:**

Q1: My main question is about the significance of Theorem 1. [1] already proved that  transformers can implement ridge regression on nonlinear dynamical systems, then what new message does Theorem 1 bring compared with [1]?

Q2: Theorem 1 used the algorithm unrolling technique, what's the benefit of this approach compared with the conventional gradient descent upadate in [1] [2] [3] [4] etc.?

Q3: While Theorem 2 suggests that a single-layer Transformer is theoretically insufficient for ICL of linear dynamical systems, could this limitation be empirically verified? Additionally, although determining the exact number of layers required remains challenging from a theoretical perspective, an empirical investigation could provide valuable insights.


[1] Guo, T., Hu, W., Mei, S., Wang, H., Xiong, C., Savarese, S., & Bai, Y. (2023). How do transformers learn in-context beyond simple functions? a case study on learning with representations. arXiv preprint arXiv:2310.10616.

[2] Von Oswald, J., Niklasson, E., Randazzo, E., Sacramento, J., Mordvintsev, A., Zhmoginov, A., & Vladymyrov, M. (2023, July). Transformers learn in-context by gradient descent. In International Conference on Machine Learning (pp. 35151-35174). PMLR.

[3] Ahn, K., Cheng, X., Daneshmand, H., & Sra, S. (2023). Transformers learn to implement preconditioned gradient descent for in-context learning. Advances in Neural Information Processing Systems, 36, 45614-45650.

[4] Bai, Y., Chen, F., Wang, H., Xiong, C., & Mei, S. (2023). Transformers as statisticians: Provable in-context learning with in-context algorithm selection. Advances in neural information processing systems, 36, 57125-57211.

**Ethical Concerns:**

["NO or VERY MINOR ethics concerns only"]

**Final Justification:**

During the rebuttal period the authors have added experiments to showcase the inability for a one layer attention to learn dynamical systems. Also most of my questions are resolved, and the algorithm unrolling technique is essentially a generalization of the GD but appears beneficial from a theoretical point of view.

**Limitations:**

yes

**Quality:**

3

**Strengths And Weaknesses:**

## Strengths

1. The structure of the paper is clear and is generally well-written.

2. The lower bound which showcases the inability of a single-layer transformer to perform ICL on dynamical systems is interesting.

## Weaknesses

Please see questions.

---

> ### Author Rebuttal · Authors · 2025-07-30
>
> We thank the reviewer for their kind and helpful feedback. Since several reviewers cited the same weaknesses, we have the following general response:
>
> $\\textbf{Experiments:}$ We ran some experiments to validate our theoretical results. Due to the current rebuttal guidelines, we cannot show the plots from our experiments. However, we plan to add our experimental results to the revised version. Our results are as follows. We trained transformers with both linear and softmax attention to learn random linear dynamical systems in-context. For computational reasons, the loss we use is averaged over the task distribution, as opposed to the supremum loss we use for theory. For one-dimensional systems with noise variance $\sigma^2 = 0.05$ and context length $T = 500$, we present the minimum test loss as as a function of depth in the following tables:
>
> $$
> \\begin{bmatrix}
> \\textrm{Number of layers (softmax attention)} & \\textrm{Minimum loss value:} \\\\
> L=1 & 0.0011824991088360548 \\\\
> L=2 & 0.00022110833378974348 \\\\
> L=5 & 0.00012664416863117367
> \\end{bmatrix}
> $$
>
> $$
> \\begin{bmatrix}
> \\textrm{Number of layers (linear attention)} & \\textrm{Minimum loss value:} \\\\
> L=1 & 0.0007882040226832032 \\\\
> L=2 & 0.00015281644300557673 \\\\
> L=5 & 0.00011635164264589548
> \\end{bmatrix}
> $$
>
> We also have numerical experiments for multi-dimensional systems (up to $d=5$), but the results are most illustrative for one-dimensional examples. Additionally, while we are unable to produce results for depth greater than $L=5$ due to the time constraint, we plan to include experimental results for deeper transformers in the revised version. Nonetheless, we hope that our inclusion of these experimental results strengthens the overall contribution of the paper. To summarize, our plots confirm the following points:
>
> 1. There is a clear separation in the error achieved by single-layer and multi-layer transformers. This provides evidence for Theorem 2 in our paper.
> 2. Increasing the depth beyond two layers  marginally improves the error, but the most significant difference is between one and two-layer transformers.
> 3. For each choice of the number of layers $L$, the minimum loss achieved by the softmax attention transformer is comparable to the minimum loss achieved by the linear attention transformer. This suggests that the non-vanishing lower bound persists when using softmax attention. The minimum loss values are shown in the table below.
>
> $\\textbf{Choice of architecture for the lower bound:}$ Broadly, the goal of our paper is to provably demonstrate the importance of depth for in-context learning when the data is not iid. We think that the linear transformer is a natural architecture through which to study this phenomenon, since several prior works have studied the in-context learning ability of  this architecture on iid data. In particular, the recent works [1,2,3] which prove upper bounds on the loss for single-layer linear transformers on iid data, provide a direct baseline to compare with our lower bound. Currently, we are investigating  whether our upper bounds can be extended to linear transformers with finite depth, as well as the approximation error of nonlinear transformers, such as those with with softmax attention modules; however, these directions are beyond the scope of the current work.
>
> Concerning your specific questions:
>
> The main message of Theorem 1 of our paper is not only the construction of a specific transformer, but that our constructed transformer achieves a next-token prediction error bound of $O(\log(T)/T)$.  This is where much of the work of proving Theorem 1 comes from, as it requires us to synthesize our particular construction with statistical guarantees of the least-squares estimator (LSE) on noisy data, and to control the error incurred on the event where the LSE bounds fail to hold. In contrast, Theorem 1 in [4] only presents the construction of a transformer which approximates the ridge regression algorithm for dynamical systems and does not bound the next-token prediction error of their construction. In addition, the two constructions are not directly comparable because their construction applies to a wider class of nonlinear dynamical systems, but our construction requires a much simpler transformer. In particular, while [4] constructs a transformer with $L + O(\log(1/\epsilon))$ layers, 5 heads, and nonlinear MLP layer a hidden dimension of $D_{hid} = 2D + d + 10$ to implement $L$ steps of gradient descent for ridge regression to accuracy $\epsilon$, our construction requires $L + O(1)$ layers, 1 head, and a linear MLP layer with a hidden dimension of $d$ to implement the least-squares predictor with the inverse covariance approximated by $L$ steps of Richardson's iteration. This demonstrates the benefit of our algorithm unrolling technique compared to the traditional GD approach (which is a different form of algorithm unrolling): it allows us to exploit the structure of the dynamical system considered herein, thus reducing the model capacity.
> References:
>
> [1] Zhang, Ruiqi, Spencer Frei, and Peter L. Bartlett. "Trained transformers learn linear models in-context." Journal of Machine Learning Research 25.49 (2024): 1-55.
>
> [2] Ahn, Kwangjun, et al. "Transformers learn to implement preconditioned gradient descent for in-context learning." Advances in Neural Information Processing Systems 36 (2023): 45614-45650.
>
> [3] Mahankali, Arvind, Tatsunori B. Hashimoto, and Tengyu Ma. "One step of gradient descent is provably the optimal in-context learner with one layer of linear self-attention." arXiv preprint arXiv:2307.03576 (2023)
>
> [4] Guo, Tianyu, et al. "How do transformers learn in-context beyond simple functions? a case study on learning with representations." arXiv preprint arXiv:2310.10616 (2023).

---

> ### Comment · Reviewer_Q9Jh · 2025-08-05
>
> Thank you for the detailed response. Most of my questions are solved, especially regarding the comparison with existing work and the benefit of the algorithm unrolling technique, which could be a contribution itself, and I willl raise my score to 4.
>
> I do have one more question regarding the experiment though, Theorem 2 states that the loss can't be smaller than a constant, does your experiment observe a platue for the one layer attention at the end of the training epochs compared with deeper layers?

---

> > ### Author Response · Authors · 2025-08-05
> >
> > Thank you once again for carefully reading and engaging with our work. We sincerely appreciate your insightful feedback, which has led to the improvement of our paper, and for your reassessment of our contributions.
> >
> > To answer your question, in fact, we observe a plateau in the loss vs epochs plot for **both the one layer and multi-layer models.** Since there is error both due to model capacity and finite context length, it is difficult to say precisely whether the plateau in the one-layer model is a reflection of the non-vanishing lower bound derived in Theorem 2 of our paper. However, we also observe that 1) the one-layer model plateaus to a **larger loss value** than the multi-layer transformer, and 2) the one-layer model plateaus **earlier during the training procedure** than the multi-layer models. We believe that these two observations support that the plateau in the loss of the one-layer models is due to the nonvanishing lower bound of Theorem 2.

---

> > > ### Comment · Reviewer_Q9Jh · 2025-08-06
> > >
> > > Thanks for the response, this makes sense. Wish you good luck!

---

### Official Review · Reviewer_dJ25 · 2025-07-03

**Clarity:** 4
**Significance:** 2
**Originality:** 2
**Rating:** 4
**Confidence:** 3

**Summary:**

The paper studies the in-context learning capabilities of transformers on non-IID data generated by noisy linear dynamical systems. The authors establish a depth-separation phenomenon. Their primary results show that while deep transformers with logarithmic depth can achieve an error rate of O(log(T)/T), which is comparable to the least-squares estimator, a class of single-layer linear transformers is fundamentally limited, suffering from a non-vanishing approximation error of Ω(1). This highlights that for certain classes of correlated, task-dependent data, network depth is a crucial architectural requirement for effective in-context learning, a finding that contrasts with results from IID settings.

**Questions:**

- The lower bound is established for a strongly simplified single-layer model. While the authors discuss several possible generalizations (such as to multi-dimensional settings or non-identity MLPs) it remains unclear how they view the role of non-linearity in standard softmax attention. Do they expect the depth separation phenomenon to persist in the presence of such non-linearities?
- The results identify a task that a one-layer linear transformer cannot learn. Have numerical experiments been conducted (using standard architectures and training) to empirically validate the predicted Ω(1) vs O(log(T)/T) error gap between shallow and deep models, or to explore potential improvements in these bounds?

**Ethical Concerns:**

["NO or VERY MINOR ethics concerns only"]

**Final Justification:**

This paper investigates in-context learning under non-IID conditions and presents a meaningful depth-separation result in a setting that more closely reflects realistic data distributions than prior work. The theoretical analysis is rigorous and clearly presented, offering valuable insights into the role of depth in learning from correlated input sequences. However, the model is highly idealized. During the rebuttal, the authors provided numerical experiments which, although limited, offer preliminary evidence that the main claims may extend beyond the simplified setting. For these reasons, I consider this a borderline accept.

**Limitations:**

Yes

**Quality:**

3

**Strengths And Weaknesses:**

**Strengths:**
- The paper provides a significant depth-separation result for in-context learning in a non-IID setting, which is more realistic than prior work.
- The analysis is sound and well-written, offering insights into why depth may be crucial for learning from correlated data sequences.

**Weakness:**
- The key lower bound result, which establishes the failure of shallow models, is proven for a significantly simplified single-layer transformer (one-dimensional system, linear attention, and a fixed identity MLP matrix). While the authors briefly discuss potential extensions, this simplification limits the immediate generality of the claim. It remains an open question how these results would change with more complex components like non-linear MLPs or softmax attention.
- The work is purely theoretical, and it would be valuable to include numerical experiments to empirically validate the predicted performance gap between shallow and deep models, particularly within a more realistic Transformer architecture. In this context, the phrase “We also provided numerical results that confirmed…” at line 298 is unclear: could the authors clarify which results are being referred to and how they relate to the theoretical findings?

---

> ### Author Rebuttal · Authors · 2025-07-30
>
> We thank the reviewer for their kind and helpful feedback. Since several reviewers cited the same weaknesses, we have the following general response:
>
> $\\textbf{Experiments:}$ We ran some experiments to validate our theoretical results. Due to the current rebuttal guidelines, we cannot show the plots from our experiments. However, we plan to add our experimental results to the revised version. Our results are as follows. We trained transformers with both linear and softmax attention to learn random linear dynamical systems in-context. For computational reasons, the loss we use is averaged over the task distribution, as opposed to the supremum loss we use for theory. For one-dimensional systems with noise variance $\sigma^2 = 0.05$ and context length $T = 500$, we present the minimum test loss as as a function of depth in the following tables:
>
> $$
> \\begin{bmatrix}
> \\textrm{Number of layers (softmax attention)} & \\textrm{Minimum loss value:} \\\\
> L=1 & 0.0011824991088360548 \\\\
> L=2 & 0.00022110833378974348 \\\\
> L=5 & 0.00012664416863117367
> \\end{bmatrix}
> $$
>
> $$
> \\begin{bmatrix}
> \\textrm{Number of layers (linear attention)} & \\textrm{Minimum loss value:} \\\\
> L=1 & 0.0007882040226832032 \\\\
> L=2 & 0.00015281644300557673 \\\\
> L=5 & 0.00011635164264589548
> \\end{bmatrix}
> $$
>
> We also have numerical experiments for multi-dimensional systems (up to $d=5$), but the results are most illustrative for one-dimensional examples. Additionally, while we are unable to produce results for depth greater than $L=5$ due to the time constraint, we plan to include experimental results for deeper transformers in the revised version. Nonetheless, we hope that our inclusion of these experimental results strengthens the overall contribution of the paper. To summarize, our plots confirm the following points:
>
> 1. There is a clear separation in the error achieved by single-layer and multi-layer transformers. This provides evidence for Theorem 2 in our paper.
> 2. Increasing the depth beyond two layers  marginally improves the error, but the most significant difference is between one and two-layer transformers.
> 3. For each choice of the number of layers $L$, the minimum loss achieved by the softmax attention transformer is comparable to the minimum loss achieved by the linear attention transformer. This suggests that the non-vanishing lower bound persists when using softmax attention. The minimum loss values are shown in the table below.
>
> $\\textbf{Choice of architecture for the lower bound:}$ Broadly, the goal of our paper is to provably demonstrate the importance of depth for in-context learning when the data is not iid. We think that the linear transformer is a natural architecture through which to study this phenomenon, since several prior works have studied the in-context learning ability of  this architecture on iid data. In particular, the recent works [1,2,3] which prove upper bounds on the loss for single-layer linear transformers on iid data, provide a direct baseline to compare with our lower bound. Currently, we are investigating  whether our upper bounds can be extended to linear transformers with finite depth, as well as the approximation error of nonlinear transformers, such as those with with softmax attention modules; however, these directions are beyond the scope of the current work.
>
> Concerning your specific questions, we have the following comments:
>
> 1. Extending our results to nonlinear transformers is an ongoing direction. Our recent experiments suggest that the depth separation observed in Theorem 2 also holds for transformers with softmax attention. However, it is technically challenging to compute the limiting loss as we did for the linear case, because it involves computing moments softmax vectors of the linear dynamical system. It therefore seems that the softmax case may require a different analytical approach to the problem.
> 2. While our experiments do confirm a depth separation, it is difficult to validate these exact rates because they are asymptotic results. In addition, the optimal rate for in-context learning linear dynamical systems is unknown. The $\frac{\log(T)}{T}$ rate is optimal (up to the log factor) for estimating the matrix $W$ from the dynamical system trajectory, but it may not be optimal for our in-context learning problem, namely, predicting the next token from the history $x_1, \dots, x_T$.
>
> References:
> [1] Zhang, Ruiqi, Spencer Frei, and Peter L. Bartlett. "Trained transformers learn linear models in-context." Journal of Machine Learning Research 25.49 (2024): 1-55.
> [2] Ahn, Kwangjun, et al. "Transformers learn to implement preconditioned gradient descent for in-context learning." Advances in Neural Information Processing Systems 36 (2023): 45614-45650.
> [3] Mahankali, Arvind, Tatsunori B. Hashimoto, and Tengyu Ma. "One step of gradient descent is provably the optimal in-context learner with one layer of linear self-attention." arXiv preprint arXiv:2307.03576 (2023).

---

> > ### Comment · Reviewer_dJ25 · 2025-08-05
> >
> > I would like to thank the authors for the clarifications provided during the rebuttal and for the additional numerical experiments. Although the explored model depth is relatively limited, I understand the time constraints. Overall, the results suggest that the phenomena analyzed in the paper may extend to a broader setting. I see these additions as a valuable motivation for further developing the idealized theoretical analysis already presented in the paper. For this reason, I will raise my score to 4.

---

> > > ### Author Response · Authors · 2025-08-05
> > >
> > > Thank you for your invaluable feedback, which has led to improvements to our paper, and for your positive assessment of our work. We are pleased to have addressed your concerns.

---

### Note · Authors · 2025-08-12

We appreciate the reviewers and AC for facilitating a productive discussion during the rebuttal period. We believe our paper has benefitted greatly from the discussions with reviewers. In particular, their feedback led us to add several experimental results which validate our theoretical findings. It also helped us to improve the clarity of our work and to emphasize the novelty and significance of our results.

---

### Decision · Program_Chairs · 2025-09-17

**Decision:**

Accept (poster)

**Comment:**

This paper investigates the in-context learning capabilities of deep linear transformers when data sequences are generated by a stable dynamical system. The authors show that if the transformer depth scales logarithmically with the sequence length T, then the test L2​ loss decreases at the rate of log(T)/T, indicating vanishing approximation error. In contrast, in a simplified setting, they prove that a single-layer transformer cannot perform in-context learning.

The paper received four reviews, all of which recognize its significant contribution in clarifying the fundamental strengths and limitations of linear transformers for in-context learning. The AC concurs with this assessment, noting that the paper introduces several new ideas and results, some with promising potential for extension to more general, nonlinear models.

That said, the reviewers also identified certain drawbacks. A major concern was the limitation of Theorem 2 in constructing a linear transformer that maintains non-vanishing error. They pointed out that the construction in Theorem 2 is considerably more restricted than the one in Theorem 1. In response, the authors provided experiments suggesting that the conclusions of Theorem 2 also hold in the broader setting of Theorem 1 (at least empirically). Another point raised was the lack of analysis on the optimization aspect of transformers. The authors clarified that optimization lies beyond the scope of this work, a position with which I agree.

Overall, this is a solid and meaningful contribution.